# Conserved protein Pir2ARS2 mediates gene repression through cryptic introns in lncRNAs

Gobi Thillainadesan[1], Hua Xiao[1], Sahana Holla[1], Jothy Dhakshnamoorthy[1], Lisa M. Miller Jenkins[2], David Wheeler[1] & Shiv I. S. Grewal [1✉]

Long non-coding RNAs (lncRNAs) are components of epigenetic control mechanisms that ensure appropriate and timely gene expression. The functions of lncRNAs are often mediated through associated gene regulatory activities, but how lncRNAs are distinguished from other RNAs and recruit effector complexes is unclear. Here, we utilize the fission yeast *Schizosaccharomyces pombe* to investigate how lncRNAs engage silencing activities to regulate gene expression in cis. We find that invasion of lncRNA transcription into the downstream gene body incorporates a cryptic intron required for repression of that gene. Our analyses show that lncRNAs containing cryptic introns are targeted by the conserved Pir2ARS2 protein in association with splicing factors, which recruit RNA processing and chromatin-modifying activities involved in gene silencing. Pir2 and splicing machinery are broadly required for gene repression. Our finding that human ARS2 also interacts with splicing factors suggests a conserved mechanism mediates gene repression through cryptic introns within lncRNAs.

[1] Laboratory of Biochemistry and Molecular Biology, National Cancer Institute, National Institutes of Health, Bethesda, MD 20892, USA. [2] Laboratory of Cell Biology, National Cancer Institute, National Institutes of Health, Bethesda, MD 20892, USA. ✉email: grewals@mail.nih.gov

ncRNAs dynamically regulate gene expression during development and in response to environmental conditions[1–4]. Defects in gene regulation by lncRNAs are frequently linked to diseases including cancer[1,2]. In many cases, lncRNAs govern gene expression by directing chromatin-modifying enzymes and other factors[2]. This function of lncRNAs is mediated via their associated proteins, but the mechanisms by which the lncRNAs selectively engage gene regulatory activities have remained largely unknown.

*S. pombe* is a powerful genetic model system for studying lncRNAs and their roles in the regulation of gene expression. In addition to numerous annotated lncRNAs, several RNA processing factors that are missing in budding yeast are conserved from *S. pombe* to higher eukaryotes. Many lncRNAs control gene expression in response to environmental and developmental signals[5–10], including *cis*-acting lncRNAs that regulate the expression of nearby genes. Examples of regulatory lncRNAs include the *prt* lncRNA that represses the acid phosphatase *pho1* gene in the presence of phosphate, and the *nam1* lncRNA that silences the mitogen-activated protein kinase *byr2* gene essential for sexual differentiation[6,7,11]. Transcription termination and degradation of the lncRNAs prevents them from invading and repressing downstream genes[7,11–14]. However, under specific growth conditions, readthrough transcription of lncRNAs leads to repression of downstream genes[15]. Underscoring a direct role, cells defective in lncRNA production show de-repression of target genes[6–8,11,12]. Although these and other lncRNAs play a critical role in mediating gene repression, the exact mechanism is not understood.

RNA processing factors that process diverse RNA species have been implicated in both posttranscriptional and transcriptional silencing[16]. RNAi machinery processes transcripts into small RNAs (siRNAs), but is also critical for targeting chromatin-modifying activities, such as factors involved in heterochromatin assembly[17,18]. The components of the RNAi pathway include the RNA-induced transcriptional silencing complex (RITS: Ago1, Chp1, and Tas3), the RNA-directed RNA polymerase complex (RDRC: Cid12, Hrr1 and Rdp1), and Dicer (Dcr1)[17–21]. In addition to playing a prominent role in processing centromeric repeat transcripts, RNAi targets various other loci, including retrotransposons, sexual differentiation genes, and genes encoding transmembrane proteins[22].

Additionally, *S. pombe* contains conserved machinery that promotes degradation of transcripts by the $3' \rightarrow 5'$ exonuclease Rrp6[6,23,24]. MTREC (Mtl1-Red1 core) is composed of the Mtr4-like RNA helicase Mtl1 and the zinc finger protein Red1 and serves as the molecular hub of an RNA processing network[6,25] related to NEXT and PAXT in mammals[26]. MTREC and its associated factors preferentially target transcripts containing hexameric DSR (determinant of selective removal) elements, which are bound by a YTH family RNA-binding protein Mmi1[23,27]. Mmi1 physically interacts with the Erh1 protein to form a complex referred to as EMC (Erh1-Mmi1 Complex). EMC recruits MTREC to meiotic genes to prevent their untimely expression during vegetative growth, in addition to targeting *cis*-acting lncRNAs including *prt* and *nam1*[6,7,12,28,29]. Mmi1 also mediates recruitment of the cleavage and polyadenylation factor (CPF) complex, which acts together with Rrp6 to trigger transcription termination and degradation of lncRNAs, thus preventing them from invading and repressing downstream genes[7,11–14]. Despite these studies, a major unanswered question is how lncRNAs mediate gene repression.

In this study, we demonstrate that besides MTREC, *cis*-acting lncRNAs show enrichment of the highly conserved Pir2 protein (ARS2 in mammals). Remarkably, lncRNAs contain cryptic introns that provide a scaffold for splicing factors and Pir2, which

are required for lncRNA-mediated gene repression. Our analyses show that the Pir2-splicing-machinery recruits silencing effector complexes to aid in the repression of target gene loci. We also find that ARS2 associates with splicing factors in human cells, suggesting that human ARS2 functions similarly to connect regulatory RNAs to gene silencing activities.

## Results

**Pir2 is required for lncRNA-mediated repression.** We investigated if MTREC and its associated factors, including the Pir2[ARS2] protein[6,25], are required for repression of *pho1* and *byr2* by lncRNA. Pir2[ARS2] is an essential protein implicated in various aspects of RNA metabolism[29,30]. Loss of the MTREC subunit Red1 resulted in the accumulation of longer readthrough transcripts (referred to as *prt-L* and *nam1-L*) (Fig. 1a), as was also observed in *mmi1Δ* and *rrp6Δ* cells (Fig. 1a and Supplementary Fig. 1a)[6,7,11]. By contrast, a mutation in *pir2* (*pir2-1*)[29] did not affect the levels of *prt* and *nam1* lncRNAs (Fig. 1b). Surprisingly, *pir2-1* showed a drastic upregulation of *pho1* and *byr2* genes as compared to wild-type (WT) (Fig. 1b), similar to the effect observed upon deletion of the lncRNA (Supplementary Fig. 1b)[6,7,11]. Chromatin immunoprecipitation followed by sequencing (ChIP-seq) confirmed Pir2 enrichment at lncRNAs, including *prt* and *nam1* (Fig. 1c and Supplementary Fig. 1c). At *nam1*, Pir2 enrichment encompassed the upstream *mlo3* locus. Moreover, RNA immunoprecipitation sequencing analysis (RIP-seq) showed that Pir2 binds to the lncRNAs (Fig. 1d and Supplementary Fig. 1d). Consistently, deletion of *prt* abolished Pir2 localization at the target locus (Fig. 1e). Together, these results suggest that lncRNAs recruit Pir2 to repress their downstream genes. Supporting the function of Pir2 and lncRNA in the same pathway, we found no additive effect on *pho1* expression in the *pir2-1 prtΔ* double mutant when compared to the effect in the single mutants (Fig. 1e).

The requirement for Pir2 in mediating the repressive effects of lncRNAs is a highly significant finding. We asked if Pir2 is also required for the repression of *byr2* that is observed upon the accumulation of *nam1* lncRNA in cells lacking Rrp6. Since *byr2* is required for meiotic induction, cells lacking Rrp6 are defective in sporulation (Fig. 1f)[11]. Remarkably, entry into meiosis and sporulation efficiency were restored in *pir2-1 rrp6Δ* cells (Fig. 1f). Similar results were obtained from a qualitative assay in which iodine vapor stains the starch-like compound produced by cells undergoing meiosis a dark brown color. Whereas *rrp6Δ* cells that are defective in meiotic induction due to *byr2* repression stained yellow, the *rrp6Δ pir2-1* double mutant colonies stained dark brown (Fig. 1f). These results support a role for Pir2 in mediating repression of *byr2* by *nam1* lncRNA.

**Pir2-CBC and splicing factors mediate gene repression.** We next tested whether Pir2-associated factors are also required for lncRNA-mediated gene repression. Consistent with co-purification of Pir2[ARS2] with the cap-binding complex (CBC) in *S. pombe* and mammals[25,30–32], Pir2 co-immunoprecipitated (co-IP) with CBC components (Supplementary Fig. 1e). Moreover, CBC co-fractionated with Pir2 in glycerol gradient analyses (Supplementary Fig. 2). To test whether CBC is required for Pir2-mediated gene repression we constructed a partial loss-of-function mutant allele of the *cbc1* gene (*cbc1-1*), which encodes an essential subunit of CBC. The lncRNA-mediated repression of *pho1* was impaired in *cbc1-1* cells (Supplementary Fig. 1e), suggesting that Pir2 likely acts together with CBC to promote gene repression.

In addition to CBC, Pir2-purified fractions also contain a subset of splicing factors including Cwf10 (EFTUD2 in human), which is a subunit of the U5 small nuclear ribonucleic particle[33],

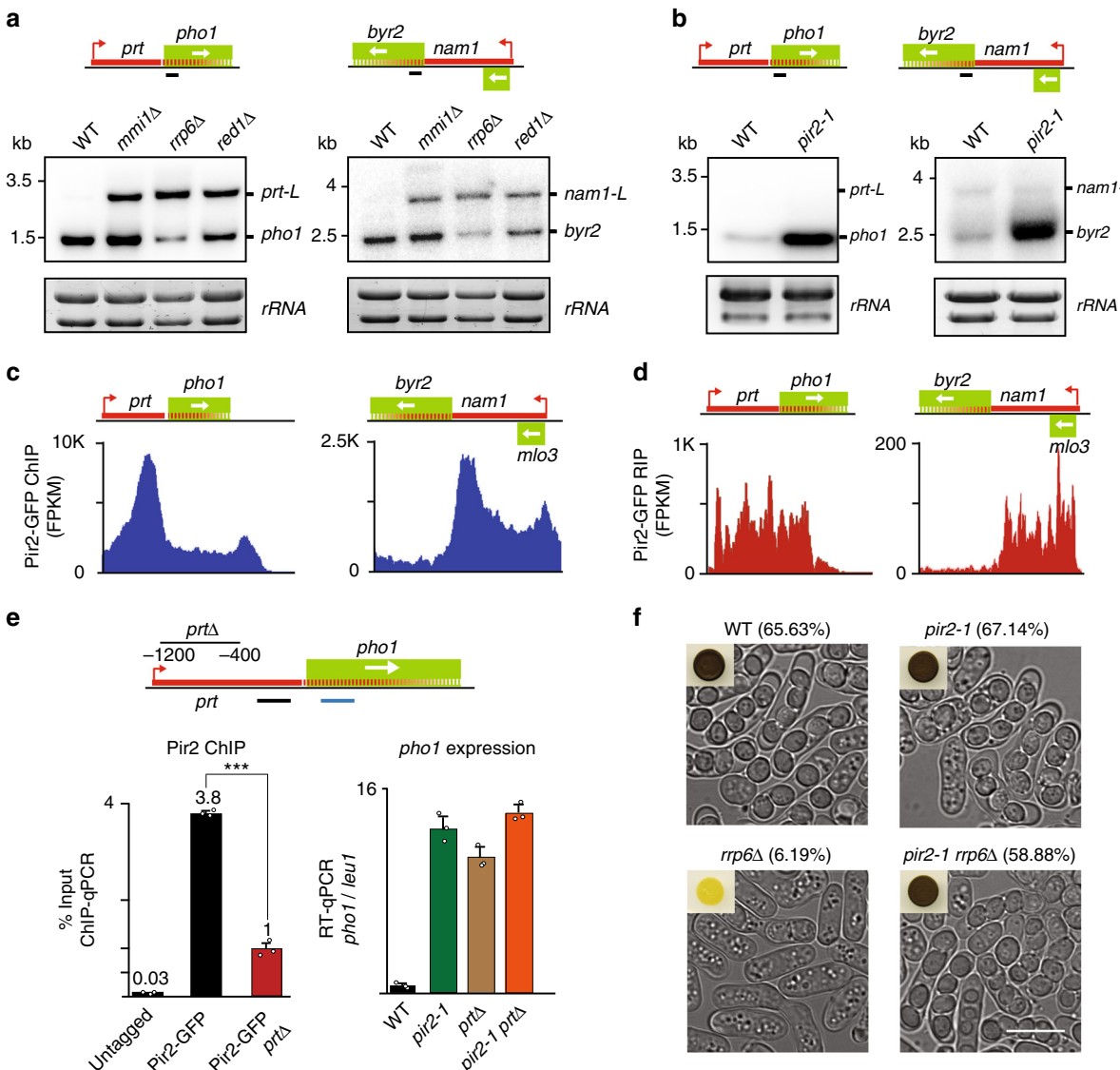

**Fig. 1 Pir2 is required for lncRNA-mediated repression of neighboring genes. a**, **b** Northern blot analysis of transcripts produced from the *pho1* and *byr2* loci. The black line indicates the position of the radioactive probe. Ribosomal RNA was used as a loading control. Cells were grown in YEA medium. Note that longer exposures were used to detect *prt-L* and *nam1-L* transcripts in (**a**). As a result, *pho1* and *byr2* bands in WT lanes are darker in (**a**) as compared to (**b**). **c** ChIP-seq analysis of Pir2-GFP enrichment at *pho1* and *byr2* loci. Source data are provided as a Source data file. **d** RIP-seq analysis of Pir2-GFP at *pho1* and *byr2* loci. **e** ChIP-qPCR analysis of Pir2-GFP (left panel). The region deleted from the *prt* lncRNA (*prtΔ*) is indicated. The amplified region is indicated by a black line. Data are presented as mean values ± SD for n = 3 biologically independent samples. Student's t-test (two-tailed) was used to calculate p-value. Between Pir2-GFP and Pir2-GFP *prtΔ*, p = 0.0009 (***p < 0.001). RT-qPCR analysis of the *pho1* gene (right panel). Transcript levels were normalized to the *leu1* control. The amplified region is indicated by the blue line. Data are presented as mean values ± SD for n = 3 biologically independent samples. Mean data distribution is represented by white circles. **f** Representative DIC images of homothallic (h⁹⁰) WT and mutant strains. The sporulation frequency is indicated as a percentage (n > 1000). Insets show colonies stained with iodine vapor. The white scale bar represents 10 μm.

and Cwf21 (SRRM2 in humans)[25]. Biochemical analyses showed that Pir2 indeed forms a complex with splicing factors. We confirmed their association by co-IP (Fig. 2a) and also found that Pir2 co-eluted with a subfraction of Cwf10 in a glycerol gradient (Fig. 2b). Cwf10 eluted in two major fractions, indicating the presence of a smaller complex containing Pir2 and a second larger complex likely representing the active spliceosome. To confirm this, we determined the elution profiles of the splicing protein Cdc5, a core component of the active spliceosome (part of the Nineteen complex; NTC), and the associated factor Spp42 (Prp8)[34–36]. Indeed, Cdc5 and Spp42 eluted as a larger complex corresponding to the active spliceosome and were not found in the smaller fraction with Pir2 (Supplementary Fig. 2).

Interestingly, our glycerol gradient analysis showed exclusive co-elution of Cwf21 and Pir2. Together, these results suggest that Pir2 forms a complex with splicing factors that are not part of the active spliceosome.

To determine whether Pir2 acts together with splicing factors to promote gene repression by lncRNAs, we performed northern blot analysis. A significant increase in the level of both *pho1* and *byr2* mRNAs in *cwf10-1* as compared to WT confirmed that the splicing machinery indeed affects the expression of genes repressed by lncRNAs (Fig. 2c). We then performed epistasis analysis to test if Pir2 and the splicing machinery are components of the same silencing pathway. We found no cumulative increase in the expression of genes repressed by lncRNA in the *pir2-1*

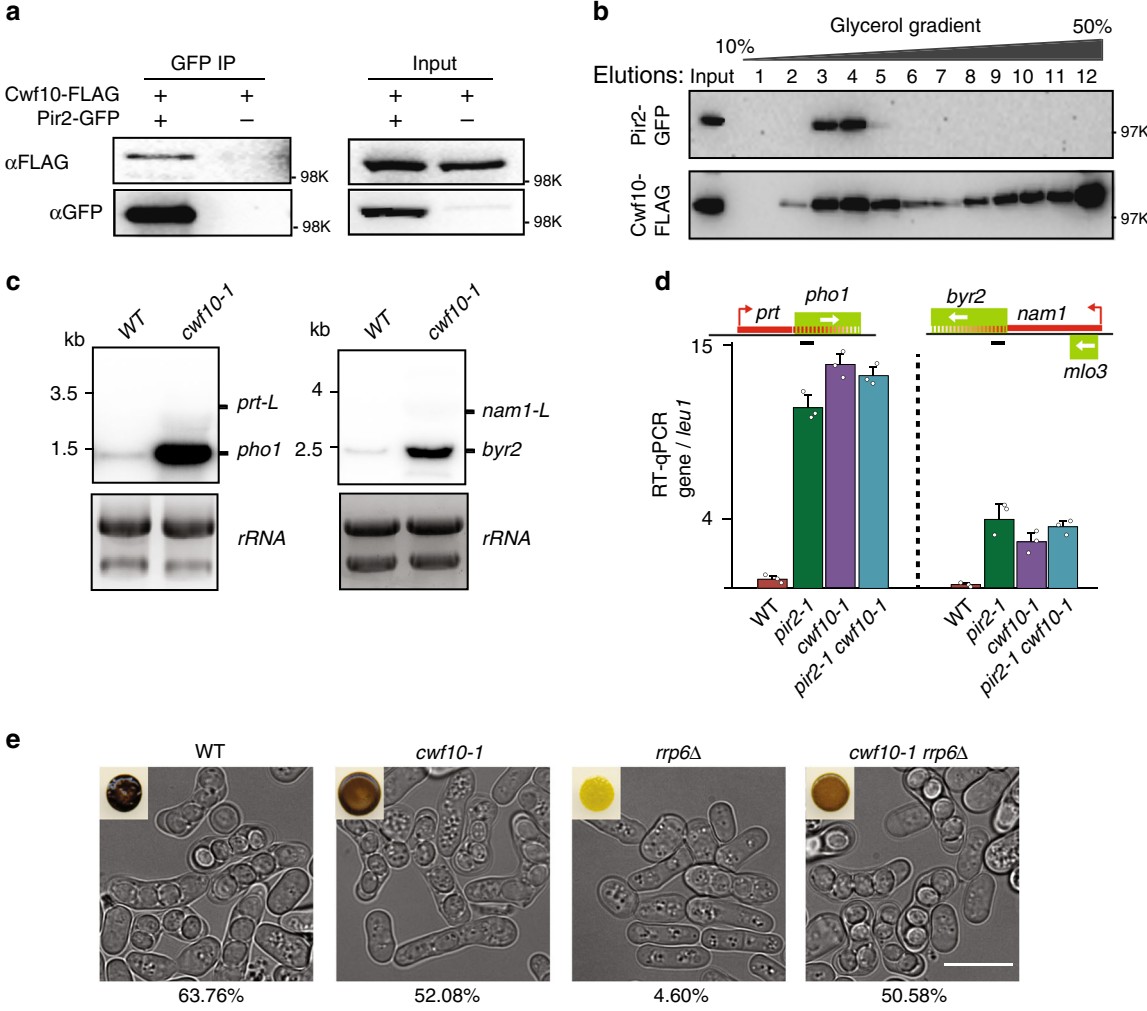

**Fig. 2 Splicing machinery bound to Pir2 mediate gene repression by lncRNA. a** Co-IP analysis of Pir2-GFP with Cwf10-FLAG proteins. Source data are provided as a Source data file. **b** Glycerol gradient analysis. Elution fractions are indicated above the gel. Source data are provided as a Source data file. **c** Northern blot analysis of transcripts produced from the *pho1* and *byr2* loci. Probes are described in Fig. 1a. Source data are provided as a Source data file. Cells were grown in YEA medium. **d** RT-qPCR analysis of *pho1* and *byr2* expression, normalized to *leu1*. The amplified regions are indicated by the black lines. Data are presented as mean values ± SD for *n* = 3 biologically independent samples. Mean data distribution is represented by white circles. **e** Representative DIC images of homothallic (h^90) WT and mutant strains. The sporulation frequency is indicated as a percentage (*n* > 1000). Insets show colonies stained with iodine vapor. Indicated white scale bar represents 10 µm.

*cwf10-1* double mutant as compared to the single mutants (Fig. 2d). Importantly, *cwf10-1* rescued the sporulation defect observed in *rrp6Δ* caused by the silencing of the *byr2* gene by *nam1* lncRNA (Fig. 2e), similar to *pir2-1* (Fig. 1f). These results confirm the biological significance of Pir2 association with splicing machinery and show that these factors collaborate to promote gene repression by lncRNAs.

**LncRNA-mediated repression requires a cryptic intron**. We considered that specific features of lncRNAs may be critical for gene repression by Pir2 and splicing machinery. Considering the involvement of splicing machinery, we searched for introns in the loci controlled by the lncRNAs. Despite the absence of annotated introns, examination of RNA-seq data from *pir2-1* and *pir2-1 rrp6Δ* cells revealed "cryptic" introns, which contain consensus splice sites but are inefficiently spliced[6], that map to the *pho1* and *byr2* loci (Fig. 3a). The detection of introns in these mutant cells likely reflects kinetic competition between splicing machinery and RNA processing factors. In cells lacking Pir2 and other factors such as Rrp6, defects in RNA degradation shift the balance in

favor of splicing machinery, ultimately leading to splicing of cryptic introns. The region upstream of *nam1* that showed Pir2 enrichment (Fig. 1c) also contained a cryptic intron (Fig. 3a). However, since splicing of the cryptic introns would disrupt the ORF, the possible biological significance of these introns was unclear. To address this, we generated two independent mutant strains containing deletions of the 5' and 3' splice sites of the cryptic intron within *pho1* (Fig. 3b). Remarkably, strains carrying splice site mutations showed significant upregulation of the *pho1* transcript (Fig. 3c), similar to the effect observed in *pir2-1, cwf10-1* and *prtΔ* (Figs. 1b, 2c and Supplementary Fig. 1b). Importantly, splice site mutations affected target gene silencing but not the level of *prt* lncRNA (Fig. 3c), analogous to the results obtained with *cwf10-1* or *pir2-1* mutants (Fig. 1b). This effect is distinct from the changes observed upon deletion of other known Mmi1 binding sites that contain introns[28].

To determine if Pir2 and the cryptic intron act together to maintain gene repression, we performed epistasis analysis. Combining *pir2-1* with the mutant cryptic intron allele did not result in further accumulation of *pho1* transcripts when compared to the single mutants, suggesting that Pir2 acts through the

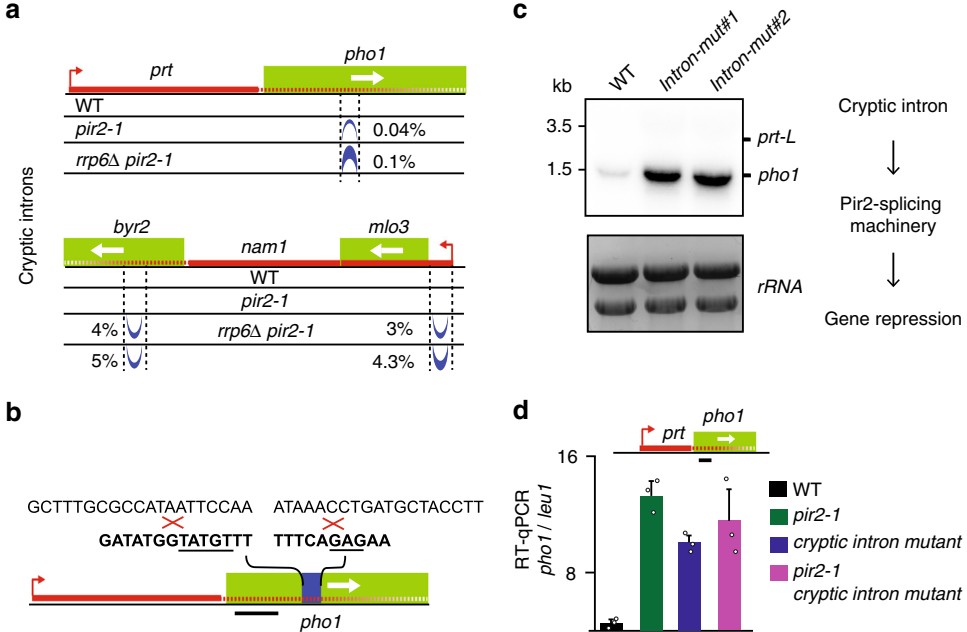

**Fig. 3 The cryptic intron within lncRNA promotes downstream gene repression. a** RNA-seq TopHat splice junctions showing cryptic introns. Numbers indicate the percentage of reads spliced. **b** Schematic showing splice site mutations in the cryptic intron of *prt-pho1*. The deleted sequence is in bold and the inserted sequence is shown above. 5′ and 3′ splice sites are underlined. **c** Northern blot analysis of transcripts produced from the *pho1* gene in WT and strains carrying the cryptic intron splice site mutations. The probe is shown in (**b**) as a black line. Source data are provided as a Source data file. Cells were grown in YEA medium. **d** RT-qPCR analysis of *pho1* expression, normalized to *leu1*. The amplified regions are indicated by the black line. Data are presented as mean values ± SD for *n* = 3 biologically independent samples. Mean data distribution is represented by white circles.

cryptic intron to promote repression of *pho1* (Fig. 3d). Together with the requirement for lncRNA to invade into the gene body, these results implicate the inclusion of the cryptic intron in the regulatory RNAs as an essential element for repression via a mechanism involving Pir2 and splicing machinery.

**Pir2 and splicing machinery collaborate genome-wide.** Cooperation between Pir2 and Cwf10 might represent a common strategy employed at other loci. Comparison of the expression profiles of *pir2-1* and *cwf10-1* revealed 435 targets repressed by both Pir2 and Cwf10, accounting for more than 50% of Pir2 target transcripts (Supplementary Fig. 3a). The targets comprised 205 mRNAs and, interestingly, 230 ncRNAs (Supplementary Data 1). Amongst the loci upregulated in *pir2-1*, we detected cryptic introns in 204 transcripts (Supplementary Fig. 3b). This is likely an underestimation due to the difficulty of detecting inefficiently spliced cryptic introns. Cryptic introns were found in mRNAs and many ncRNAs that collectively show transcript upregulation as determined by comparing RNA-seq data from *pir2-1* and *cwf10-1* cells to WT (Supplementary Fig. 3c). Notably, we detected cryptic introns in transcripts arising from retrotransposon *Tf2* elements in *pir2-1* cells (Supplementary Fig. 3d). Analysis of *pir2-1* and *cwf10-1* mutants revealed that expression of *Tf2* elements increased in both mutant strains (Supplementary Fig. 3d, e), and RIP-seq analysis showed that Pir2 binds to *Tf2* transcripts (Supplementary Fig. 3f). These results establish cryptic introns as a common feature among Pir2 targets and show that Pir2 and splicing machinery collaborate to repress genes and retrotransposons.

**Pir2-splicing machinery recruit RNAi proteins for repression.** How might Pir2 trigger repression by lncRNA containing cryptic introns? Although lncRNA production is important for loading

silencing factors[7,22,37,38], the exact mechanism has remained unclear. Since Pir2 homologs are involved in RNAi[29,31,39,40] and splicing machinery is implicated in siRNA production[41–43], we examined the association of Pir2 with RNAi machinery. Co-IP analysis showed Pir2 associates with the Hrr1 subunit of RDRC (Fig. 4a). Interestingly, this interaction was impaired in the *cwf10-1* mutant, indicating that splicing factors are required for association of Pir2 with Hrr1 (Fig. 4b). We then analyzed the role of Pir2 in siRNA production in cells lacking Rrp6, which show accumulation of lncRNAs and robust repression of their target loci. We found that siRNAs, which ranged in size from 20–24 nt and mapped to lncRNAs targeting *pho1* and *byr2*, were abolished in both *pir2-1* and *cwf10-1* mutant backgrounds (Fig. 4c and Supplementary Fig. 4a, b). The remaining reads in the mutants displayed a broad length distribution consistent with degradation products (Supplementary Fig. 4b). Pir2 was also required for siRNA production at *Tf2* elements, pericentromeric repeats, and other loci (Fig. 4c, Supplementary Fig. 5 and Supplementary Data 2). Moreover, introns detected in pericentromeric repeat transcripts in RNAi mutants[6,44] could also be observed in *pir2-1* (Supplementary Fig. 5). Based on these results, we conclude that Pir2 targets RNAi machinery to lncRNAs and other transcripts containing cryptic introns.

**Cryptic intron is required for siRNA production at lncRNA.** We next wondered whether cryptic introns are required for Pir2-dependent generation of siRNAs. Mutations of the *pho1* cryptic intron splice sites in *rrp6Δ* cells abolished the production of siRNAs mapping to the entire *prt* lncRNA, including the region upstream of *pho1* (Fig. 4d). This result suggests that the cryptic intron acts as part of the *prt* lncRNA to engage RNAi machinery. Importantly, siRNAs mapping to other loci were not affected (Fig. 4d and Supplementary Fig. 5), indicating that the observed

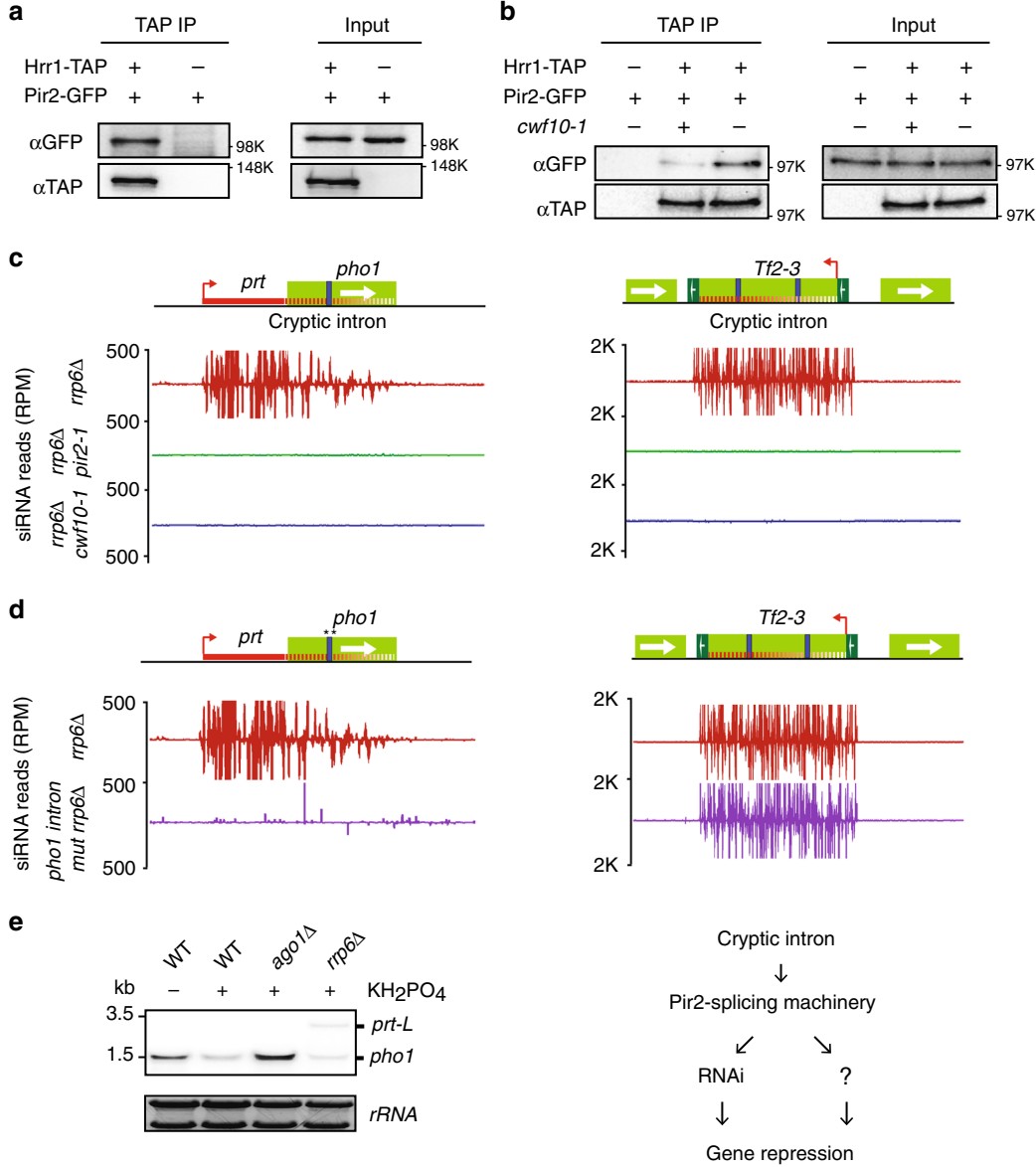

**Fig. 4 Pir2 interacts with RNAi machinery required for lncRNA-mediated gene repression. a**, **b** Co-IP analysis of Hrr1-TAP and Pir2-GFP proteins in the indicated strains. Source data are provided as a Source data file. **c** The normalized number of siRNA reads are plotted in alignment with *pho1* and *Tf2-3* loci. The signal above and below the line represent siRNAs that map to the top and bottom strands, respectively. TopHAT splice junctions of introns emerging in the strains are indicated by blue arcs. **d** Normalized number of siRNAs mapping to *pho1* and *Tf2-3* loci in *rrp6Δ* (red) and *rrp6Δ* cells carrying the splice site mutation in the cryptic intron in *prt-pho1* (purple). The cryptic intron is marked by a blue box and stars indicate mutation sites. **e** Northern blot analysis of the *pho1* gene. Cells were grown in EMM medium with or without phosphate. The radioactive probe is described in Fig. 1a. Source data are provided as a Source data file.

effect was specific to *prt-pho1*. This finding is consistent with our result showing that *pho1* is upregulated in cells carrying cryptic intron splice site mutations (Fig. 3c) and led us to examine the effects of RNAi factors on lncRNA-mediated gene repression. Cells lacking Ago1 showed a considerable increase in *pho1* transcript levels as determined by northern blot analysis (Fig. 4e), but the observed effect was weaker than in *pir2-1* or *cwf10-1*, suggesting that additional factors likely cooperate with Pir2-splicing machinery.

**Pir2 and splicing machinery target chromatin modifiers**. In addition to RNAi machinery, gene silencing by lncRNAs also requires Clr3 histone deacetylase (HDAC) and the FACT histone chaperone complex[8,38], which were detected in a Pir2-purified

fraction[30]. Therefore, we addressed whether Pir2 also recruits Clr3 and/or FACT. Clr3 and the Pob3 subunit of FACT associated with Pir2 in our biochemical analyses (Fig. 5a, b). The loss of Clr3 or Pob3 caused an increase in *pho1* transcript levels, consistent with their involvement in repression by lncRNA[8,38], but the extent of upregulation was less than in *pir2-1* (Fig. 5c). This could be due to the recruitment of multiple effectors by Pir2 to promote gene repression. To test this possibility, we compared *pho1* transcript levels in single and double mutants. Combining *pob3Δ* with other mutants resulted in severe growth defects, precluding their analysis. However, the *ago1Δ clr3Δ* double mutant showed cumulative de-repression of *pho1* (Fig. 5d). By contrast, the *pir2-1 clr3Δ* and *pir2-1 ago1Δ* double mutant strains showed no cumulative increase in transcripts as compared to the

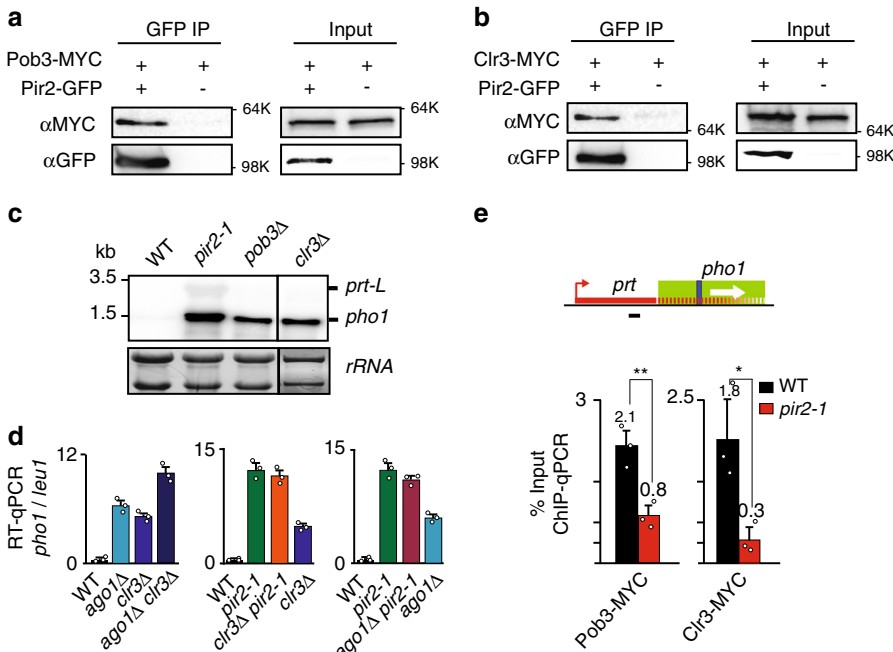

**Fig. 5 Pir2 recruits Clr3 HDAC and FACT to promote gene repression by lncRNA. a, b** Co-IP analysis of Pir2-GFP with Pob3-MYC and Clr3-MYC. Source data are provided as a Source data file. **c** Northern blot analysis of transcripts produced from the *pho1* locus. The radioactive probe is described in Fig. 1a. **d** RT-qPCR analysis of *pho1* expression, normalized to *leu1*. The amplified region is indicated by the black line. Data are presented as mean values ± SD for n = 3 biologically independent samples. Source data are provided as a Source data file. Mean data distribution is represented by white circles. **e** ChIP-qPCR of Pob3-MYC and Clr3-MYC proteins in WT (black bars) and *pir2-1* (red bars). Data are presented as mean values ± SD for n = 3 biologically independent samples. Student's *t*-test (two-tailed) was used to calculate *p*-value. Between Pob3-MYC and Pob3-MYC *pir2-1*, p = 0.008851 (**p < 0.01). Between Clr3-MYC and Clr3-MYC *pir2-1*, p = 0.008851 (*p < 0.05). Mean data distribution is represented by white circles.

single mutants, suggesting that Pir2 is epistatic to both RNAi and the Clr3 HDAC. Moreover, quantitative ChIP analyses showed enrichment of Clr3 and Pob3 at *prt-pho1* in WT cells and a reduced localization in *pir2-1* cells (Fig. 5e).

Given that lncRNAs mediate repression by chromatin modifiers and RNAi, we asked whether RNA was required to mediate interactions between Pir2 and its various interacting partners. Co-IP experiments performed in the presence of Benzonase, a DNA and RNA nuclease, revealed that except for Pob3, all other interactions with Pir2 were maintained (Supplementary Fig. 6). This suggests that Pir2 interacts with components of this pathway in an RNA-independent manner. However, RNA may have a role in promoting co-transcriptional association of Pob3 with Pir2. These results suggest that lncRNA with a cryptic intron provides a docking site for the Pir2-splicing complex, which in turn recruits multiple silencing effectors to repress gene expression (Fig. 6).

**Pir2/ARS2-splicing factor connection is conserved in humans.** The conservation of Pir2[ARS2] suggested that its functional interactions and role in lncRNA-mediated repression might be relevant to mammalian systems. We asked if ARS2 formed similar interactions in human cells using RIME (rapid immunoprecipitation mass spectrometry of endogenous proteins), which can detect transient chromatin associated co-transcriptional interactions. In addition to the known ARS2 interaction partners CBC, NEXT, and PAXT[31,32,40], RIME analysis identified additional proteins, including factors involved in nonsense-mediated decay (NMD), pre-mRNA 3'-end processing and chromatin modifiers such as FACT and HDACs (Supplementary Fig. 7a, b and Supplementary Data 3). Notably, the most abundant associating

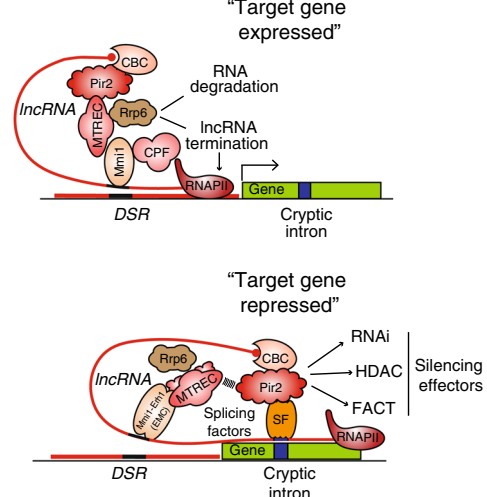

**Fig. 6 Pir2 and splicing machinery engage cryptic introns within lncRNA to promote repression.** Model depicting the switch to co-transcriptional repression through the cryptic intron. For gene expression, lncRNA is terminated and degraded upstream of target loci by CPF, MTREC, and Rrp6, which are recruited by Mmi1 bound to DSR elements. Upon changes in growth conditions, the lncRNA extends into the neighboring region containing the cryptic intron, providing a scaffold for co-transcriptional recruitment of Pir2-splicing machinery, which in turn recruits silencing effectors including RNAi, HDAC, and FACT to promote gene repression. Mmi1 acts as part of the EMC to engage MTREC and other factors that together with CBC stabilize Pir2 association with lncRNAs and effector proteins.

partners of ARS2 were splicing factors (Supplementary Fig. 7b, c). Indeed, splicing factors co-eluted with ARS2 in a glycerol gradient (Supplementary Fig. 7d), analogous to our observations in *S. pombe* (Fig. 2b). Moreover, we observed association of the human counterpart of Cwf10, EFTUD2, with ARS2, suggesting that the Pir2-splicing factor connection observed in fission yeast is conserved in humans (Supplementary Fig. 7a, e). Based on these findings, we envision that ARS2 bound to splicing machinery may function similarly to bridge regulatory RNAs to gene silencing activities.

## Discussion

Our analyses reveal that a cryptic intron within the lncRNA is a crucial element for gene repression via a pathway involving Pir2[ARS2] and splicing factors (Fig. 6). The readthrough transcription of intergenic lncRNA incorporates a cryptic intron and creates a scaffold for the co-transcriptional recruitment of the splicing machinery–Pir2 complex, which in turn engages silencing effectors. Our findings highlight a previously unrecognized role for splicing factors in engaging Pir2[ARS2] to lncRNAs to dynamically control gene expression. To this end, we find that Pir2 associates with splicing factors such as Cwf21 and Cwf10 as part of a smaller complex and is not part of the larger spliceosome complex containing the NTC components. This finding suggests that the subset of splicing factors responsible for mediating gene repression is likely distinct from the active spliceosome.

Since cryptic intron-mediated silencing occurs specifically in the context of lncRNA but not the target gene transcript, additional factors bound to lncRNA are likely involved. Other factors that bind to lncRNA might help recruit and/or stabilize Pir2 with its associated silencing effectors. In this regard, we note that loss of ERH family protein, Erh1, that associates with lncRNAs and Mmi1 as part of EMC, severely affects repression of target gene loci[29,45]. Moreover, a mutation in Mmi1 that specifically disrupts EMC assembly without affecting its termination functions impairs lncRNA-mediated repression of neighboring genes[45]. EMC bound to lncRNA may directly recruit Pir2 or may act in conjunction with other factors. Consistent with the latter possibility, EMC co-purifies with MTREC[29], which forms a complex with Pir2[6,25,30]. Therefore, MTREC, recruited by Mmi1/EMC bound to DSR elements, may act together with splicing factors engaged by the cryptic intron to promote Pir2 association with lncRNAs. In other words, the repressive effects of lncRNA require combinatorial and likely cooperative action of factors that bind to different elements embedded within regulatory RNAs (Fig. 6).

Once recruited to lncRNAs, Pir2 coordinates multiple effectors, including RNAi, to promote gene repression. In this regard, we note that Pir2 is the elusive factor that enables splicing machinery to selectively recruit RNAi to specific transcripts. RNAi processes transcripts and triggers assembly of repressive heterochromatin[16–18,22]. In addition, Pir2 promotes lncRNA-mediated recruitment of Clr3 HDAC and FACT, which increase nucleosome occupancy[37,38,46], but may also engage additional factors to enforce gene repression. Pir2 and cryptic intron-based mechanisms repress targets throughout the genome and may be conserved in higher eukaryotes. Indeed, we note that ARS2 interacts with splicing factors in human cells, and lncRNAs such as XIST implicated in X-chromosome inactivation contain inefficiently spliced introns[47,48]. Considering that defects in lncRNA-mediated gene regulation contribute to human diseases including cancer and that ARS2 is commonly mutated in various types of cancer, our findings may shed light on pathways contributing to the misexpression of genes, ultimately leading to the development of specific therapeutic strategies.

## Methods

**Cell lines, strain construction, and growth conditions**. The fission yeast strains used in this study are listed in Supplementary Table 1. Strains were generated through genetic crosses or were constructed using a PCR-based method. A DNA cassette containing a selection marker with or without epitope tag was amplified using long oligonucleotides with homology to target gene loci. The PCR product was transformed and transformants were grown on appropriate selection plates. Deletions or tagged alleles were confirmed by PCR. Cells were cultured in YEA media using standard protocols unless otherwise noted in the figures. Since *pir2*, *cbc1*, and *cwf10* are essential genes, we used partial loss of function mutant alleles. *pir2-1* and *cbc1-1* were generated using an error prone PCR method[6]. The *pir2-1* mutant allele carries two amino acid substitutions: F165L and S316P. The *cbc1-1* mutant allele contains one amino acid substitution: L119P. *cwf10-1* is a gift from R. Allshire. For the generation of cryptic intron mutants, a strain containing a *ura4+* selectable marker inserted at the *pho1* locus was transformed with a DNA fragment containing the *pho1* ORF with mutations in the splice sites. Splice site mutations were confirmed using Sanger sequencing and production of lncRNA was confirmed by RT-PCR. *pir2-1* and *cbc1-1* mutant strains were grown at 26 °C to an $OD_{600}$ 0.5 prior to shifting to 33 °C for 5 h. Cells carrying the partial loss function *cwf10-1* mutant were cultured under conditions that do not affect normal splicing[6]. For experiments indicating phosphate or no phosphate growth conditions, Edinburgh minimal media (EMM) was prepared with or without 15.5 mM sodium phosphate and 20 mM potassium phosphate. HepG2 cells were purchased from ATCC (ATCC HB-8065). Monolayers of HepG2 cells were cultured in DMEM with 10% FBS at 37 °C and 5% CO2. According to ATCC, cells were authenticated and mycoplasma tests were done using Hoechst and direct culture method followed by microscopy techniques.

**Sporulation assay**. Cells were spotted onto EMM (Fig. 1f) or Pombe minimal glutamate (PMG) medium (Fig. 2e) plates and grown at 30 °C for 3 days prior to exposure to iodine vapor. They were subsequently mounted on a 2% agarose pad for differential interference contrast (DIC) imaging using SoftWoRx V7.0 software on a DeltaVision Elite fluorescence microscope (Applied Precision, GE Healthcare) with Olympus 100×/1.40 objective. Fiji V1.0 (ImageJ, National Institutes of Health) was used for processing the images and counting the sporulation frequencies. Sporulation efficiency was monitored in more than 1000 cells from three independent isolates for each strain.

**Co-immunoprecipitations**. For co-IP experiments in fission yeast, 1 L of *S. pombe* cells was grown overnight to $OD_{600}$ 0.8 and cells were harvested by vacuum filtration. Cell pellets were flash frozen in liquid nitrogen. Cells were lysed using a CryoMill (Retsch) at a setting of 30 (frequency per second) for 1 min and repeated 3 times with 30 s intervals. Lysed cells were resuspended in lysis buffer (50 mM Tris-HCl pH8, 150 mM NaCl, 10% glycerol, 1% NP-40, 5 mM EDTA) containing Halt Protease Inhibitor Cocktail (Thermo Fisher Scientific). Lysates were incubated with either anti-GFP (Roche) bound to protein A magnetic beads (NEB) or IgG sepharose (GE) for 2 h at 4 °C. To remove DNA and RNA, proteins bound to beads were washed and treated with 50 U/ml Benzonase for 30 min. Antibody–protein complexes were eluted off the beads using 1× sample buffer and heated to 95 °C for 5 min. For western blot analysis, samples were run on a 10% Tris-glycine gel, transferred to PVDF (Thermo Fisher Scientific) and probed using anti-FLAG (M2 Sigma, F1804) at dilution 1/1000, anti-GFP (Roche) at dilution 1/1000, or anti-c-MYC (Covance, 9E10) antibodies at dilution 1/250.

For co-IP experiments in mammalian cells, human HepG2 cells were grown to 70–90% confluency and the cell nuclei were isolated in hypotonic buffer (20 mM HEPES pH 7.4, 10 mM KCl, 2 mM MgCl2, 1 mM EDTA, protease inhibitors) and lysed in lysis buffer (50 mM Tris pH 8, 0.3 M KCl, 0.5 mM EDTA/EGTA, 0.25% NP-40) by resuspending the nuclear pellet using a 20G needle. Lysates were incubated with anti-ARS2 antibody (Abcam, ab192999) for 2 h and washed 5 times with lysis buffer. Elution and western blotting were performed as described above. Anti-EFTUD2 (Abcam, ab72456) and anti-ARS2 (Abcam, ab192999) antibodies were used to probe the western blots.

**Glycerol gradient analysis**. To prepare yeast cell lysates, cells were lysed using a CryoMill (Retsch) as described above and the cell powder was resuspended in 10% glycerol buffer (20 mM Tris pH 7.5, 137 mM NaCl, 10% Glycerol, 0.5% NP40, 2 mM EDTA). For human HepG2 cells, nuclear extract was prepared as described above and the nuclei were lysed in 10% glycerol buffer by passing through a 20G needle 10 times. The lysate was resolved by loading 150 μg onto a linear 20–50% glycerol gradient prepared in an ultracentrifuge tube (Beckman, 347357). The gradients were spun in an Optima TLX-ultracentrifuge (Beckman) at 81,400 × g for 19 h. Fractions were collected by pipetting and were resolved on a 4–12% gradient gel (Invitrogen, NP0336BOX), followed by western blot analysis with ARS2 and EFTUD2 antibodies as described above. Western blotting to detect *S. pombe* proteins was performed with the following antibodies: GFP (Pir2-GFP detection; Roche, 11814460001) at dilution 1/1000, HA (Cdc5-HA and Spp42-HA detection; Biolegend, 901501) at dilution 1/1000 and FLAG (Cbc1 detection; M2 Sigma, F1804) at dilution 1/1000.

**Northern blot analysis**. For most experiments, cells were cultured in YEA medium that contains phosphate. In experiments comparing levels of *pho1* expression in the presence or absence of phosphate, EMM was used with or without 15.5 mM sodium phosphate and 20 mM potassium phosphate. Total RNA was isolated by incubating cells in hot phenol heated to 65 °C for 10 min followed by 3 additional extractions using phenol-chloroform. RNA was precipitated using the sodium-acetate-ethanol method. Northern blots were performed according to the published protocol[6]. 10 μg of RNA was resolved on a 1% formaldehyde-agarose denaturing gel and capillary transferred using NorthernMAX transfer buffer (Thermo Fisher Scientific) onto positively charged BrightStar-Plus nylon membrane (Ambion) and crosslinked using UV Stratalinker 2400 (Stratagene). The T7 in vitro transcription kit (Promega) was used to generate α-P[32]-UTP (PerkinElmer) labeled RNA probes (Supplementary Table 2) that were hybridized to the membrane overnight at 65 °C in ULTRAhyb buffer (Ambion). The membrane was exposed and scanned using a Typhoon FLA 9500 phosphor imager (GE Healthcare).

**RT-qPCR**. Total RNA was extracted as described above and treated with RQ1 DNase (Promega) followed by phenol-chloroform extraction and ethanol precipitation. Strand specific reverse transcription was performed using Revertaid Reverse Transcriptase (Thermo Fisher Scientific) with gene specific reverse primers (Supplementary Table 2) and quantified by performing qPCR with iTaq Universal SYBR Green Supermix (Bio-Rad) on the QuantStudio 3 platform (Thermo Fisher Scientific).

**ChIP and ChIP-sequencing**. ChIP experiments were performed according to the published procedure[49]. *S. pombe* cells at OD$_{600}$ 0.5 were crosslinked using 1% formaldehyde at room temperature for 20 min. Cells were spun at $2000 \times g$ for 10 min and pellets were lysed using glass beads and lysis buffer (50 mM Hepes/KOH, 140 mM NaCl, 1 mM EDTA, 1% Triton X100, 0.1% DOC plus protease inhibitors). Cell lysates were sheared using a Bioruptor-300 (Diagenode) to an approximate size of 300–600 bp. Immunoprecipitation of the protein of interest was accomplished using 50 μl of anti-c-MYC affinity gel (Sigma, A7470) or 5ug of anti-GFP antibody (Abcam, ab290). Protein A magnetic beads (NEB) were used to capture the GFP-protein–antibody complexes. Beads were washed twice with lysis buffer, twice with lysis buffer containing 0.5 M NaCl and once in TE buffer pH 8. Chromatin–antibody complex was eluted using TES buffer (50 mM Tris-HCl pH 8, 10 mM EDTA, 1% SDS) and, along with whole-cell extract (WCE) input, were de-crosslinked by heating to 65 °C overnight and purified using PCR purification columns (Qiagen). Immunoprecipitated and input DNA were assessed using iTaq-qPCR SYBR green supermix (Bio-Rad) according to the manufacturer's recommendations. Sequencing libraries were prepared using the NEBNext Ultra II DNA Library Prep Kit for Illumina according to the manufacturer's protocol and analyzed using an Agilent 4200 TapeStation system (Agilent). Sequencing was performed on the NextSeq500 platform (Illumina).

**Small RNA-sequencing**. Total RNA was isolated from a total of 4 OD$_{600}$ units of log-phase cells using the hot-phenol method as previously described above for northern blot analysis. smRNAs between 15 and 30 nucleotides were excised from 17% Urea-PAGE gels and subsequently eluted using Corning Costar Spin-X columns (Sigma Aldrich) followed by ethanol precipitation overnight. Pellets were resuspended using DEPC treated water and libraries were constructed using the NEBNext Small RNA Library for Illumina (NEB) according to the manufacturer's protocol. Qiagen MinElute PCR-purification columns were used to purify the resulting libraries which were further resolved through a 6% PAGE gel, eluted using Corning Costar Spin-X columns and ethanol precipitated. The precipitated RNA was resuspended in 1×TE buffer and analyzed as described above. The resulting libraries were sequenced on the MiSeq platform (Illumina).

**RNA-sequencing**. Total RNA was isolated as described above. rRNA was removed using the Ribo-Zero Magnetic Gold Kit for yeast (Illumina). Libraries were made using the NEBNext Ultra Directional RNA Library Prep kit for Illumina (NEB) according to the manufacturer's instructions. Libraries were analyzed and sequenced on the MiSeq platform (Illumina) as described above.

**RNA immunoprecipitation**. *S. pombe* cells expressing either GFP-tagged Pir2 (Pir2-GFP) or untagged Pir2 were grown in 50 ml of YEA medium at 30 °C to an OD$_{600}$ 0.5. Cells were then crosslinked by adding formaldehyde to a final concentration of 1% for 20 min with gentle shaking. After adding glycine to a final concentration of 0.2 M to stop crosslinking, cells were then resuspended in lysis buffer (50 mM HEPES pH 7.9, 140 mM NaCl, 1 mM EDTA, 10% glycerol, 0.5% NP-40, 0.25% Triton X-100) supplemented with complete EDTA-free proteinase inhibitor cocktail (Roche) and RNase inhibitors (Thermo Fisher Scientific, am2694) and lysed by bead beating. The lysate was sheared by sonication using a Bioruptor-300 (Diagenode). 1.5% of the original lysate for RNA preparation was set aside as input. The rest of the lysate was pre-cleared using 0.9 mg of pre-washed protein G Dynabeads (Invitrogen, 10004D) at 4 °C for 1 h. Anti-GFP antibody (Abcam, ab290) was added to the lysate and incubated with gentle rotation at 4 °C overnight. Antibody–protein complexes were captured using 1.2 mg of protein G Dynabeads for 2 h at 4 °C. Beads were washed once in 900 μl of lysis buffer, once in 900 μl of lysis buffer with 300 mM NaCl, once in 900 μl of LiCl buffer (50 mM HEPES pH7.5, 250 mM LiCl, 0.5% NP-40, 0.1% sodium deoxycholate), and once in 900 μl of TE pH 7. Beads were eluted twice in 75 μl of RIP elution buffer (50 mM Tris pH 8, 10 mM EDTA, 300 mM NaCl, 1% SDS) at 37 °C for 10 min. To the 50 μl input samples, 100 μl of RIP elution buffer was added to a final volume of 150 μl. Then, 20 μg of proteinase K (Thermo Fisher Scientific, am2548) was added to both IP and input samples, and the mixtures were incubated at 37 °C for 1 h and then at 65 °C for 1 h to de-crosslink. The samples were then extracted once with phenol-chloroform and once with chloroform, precipitated with ethanol, and resuspended in 80 μl of DEPC treated water. The samples were further treated with 20 units of RNase-free DNase I (Thermo Fisher Scientific, am2222) for 1 h at 37 °C, extracted with phenol-chloroform, and ethanol precipitated as above. IP and input samples were resuspended in 30 μl and 100 μl water, respectively. RNA concentrations were determined using a Qubit fluorometer (Thermo Fisher Scientific). Libraries were directly generated from the IP sample. The total input sample was first subjected to ribosomal RNA removal before library preparation. The libraries for IP and input samples were prepared and analyzed as described above for RNA-seq. The libraries were sequenced using the NextSeq 500 platform (Illumina).

**RIME**. Rapid immunoprecipitation mass spectrometry of endogenous proteins technique was performed according to the published procedure[50]. Monolayers of 1E8 HepG2 cells were crosslinked with 0.8% formaldehyde for 8 min at room temperature. Crosslinking was quenched with 0.1 M (final) glycine and cell nuclei were isolated using buffer LB1 (50 mM HEPES-KOH pH 7.5, 140 mM NaCl, 1 mM EDTA, 10% glycerol, 0.5% NP-40/Igepal CA-630 and 0.25% Triton X-100). Nuclei were washed with buffer LB2 (10 mM Tris-HCL pH 8.0, 200 mM NaCl, 1 mM EDTA and 0.5 mM EGTA) and lysed with buffer LB3 (10 mM Tris-HCl pH 8.0, 100 mM NaCl, 1 mM EDTA, 0.5 mM EGTA, 0.1% (wt/vol) sodium deoxycholate and 0.5% (vol/vol) *N*-lauroylsarcosine). A Bioruptor-300 (Diagenode) was used to sonicate the cells to yield DNA fragments of size range 200-500 bp followed by full speed centrifugation at 4 °C for 10 min. The resulting supernatant was incubated with 10ug of anti-ARS2 antibody (Abcam, ab192999) or 10ug of rabbit IgG mock (Abcam, ab37415) bound to protein A magnetic beads (NEB) for 4 h at 4 °C with slow rotation. Beads were washed 10 times with RIPA buffer (50 mM HEPES pH 7.6, 1 mM EDTA, 0.7% (wt/vol) sodium deoxycholate, 1% NP-40 and 0.5 M LiCl) and twice with 100 mM ammonium hydrogen carbonate buffer. The protein–antibody complex was trypsinized overnight while on the beads and the peptides were extracted using Pierce C18 Spin Columns (Thermo Fisher Scientific). Extracted peptides were dried using a Vacufuge (Eppendorf).

For mass spectrometry analysis, tryptic peptides were trapped on a trapping column and separated on a 75 μm × 15 cm, 2 μm Acclaim PepMap reverse phase column (Thermo Scientific) using an UltiMate 3000 RSLCnano HPLC (Thermo Scientific). Peptides were separated at a flow rate of 300 nL/min followed by online analysis by tandem mass spectrometry using a Thermo Orbitrap Fusion mass spectrometer. Peptides were eluted into the mass spectrometer using a linear gradient from 96% mobile phase A (0.1% formic acid in water) to 55% mobile phase B (0.1% formic acid in acetonitrile) over 210 min. Parent full-scan mass spectra were collected in the Orbitrap mass analyzer set to acquire data at 120,000 FWHM resolution; ions were then isolated in the quadrupole mass filter, fragmented within the HCD cell (HCD normalized energy 32%, stepped ± 3%), and the product ions analyzed in the ion trap. Proteome Discoverer 2.2 (Thermo) was used to search the data against human proteins from the UniProt database (downloaded January 2017) using SequestHT v1.17. The search was limited to tryptic peptides, with maximally two missed cleavages allowed. Cysteine carbamidomethylation was set as a fixed modification and methionine oxidation set as a variable modification. The precursor mass tolerance was 10 ppm, and the fragment mass tolerance was 0.6 Da. The Percolator node was used to score and rank peptide matches using a 1% false discovery rate. Protein false discovery rate was set at 1% and a minimum of 1 unique peptide required for protein identification. The mass spectrometry proteomics data have been deposited to the ProteomeXchange Consortium via the PRIDE partner repository with the dataset identifier PXD018373 and 10.6019/PXD018373.

**Bioinformatic analyses**. ChIP sequences were aligned to the genome using Bowtie2[51]. RIP and total RNA sequencing were aligned to the genome using TopHat2 and for RNA-seq, FPKM was calculated using cufflinks[52]. Normalized (RPKM) bedgraph files for aligned BAMs were generated using Deeptools[53]. Introns were assessed using TopHat2 junctions function and mutant cryptic introns were derived by subtracting annotated and WT introns. Unlike annotated introns that are spliced in majority of reads, cryptic introns are inefficiently spliced in less than 5% of the total reads spanning the region containing the intron. Cryptic intron percentage was calculated by taking the number of reads that contained a split read, indicating a spliced event, and dividing that number by the total number of reads that overlap the intron junction. For RIP-seq, the signal obtained from the untagged strain was subtracted from Pir2-GFP signal. siRNA sequencing reads were aligned using Novoalign V2 (Novocraft) and aligned reads were processed to retain 21–24 nucleotide reads and generate SGR files using a python script. The resulting file was normalized to million mapped reads. The *S. pombe* genome version used in this study is ASM294v2. For RIME (Supplementary Fig. 7), *p*-values for peptide detection were estimated from the observed distribution of log2

(#PSMs-Ars2/#PSMs-Ig-Control). This work employed the computational resources of the NIH HPC Biowulf cluster (http://hpc.nih.gov).

**Statistics and reproducibility**. Experiments for Figs. 1a, b, e, f; 2a–c, e; 4a, b, e and Supplementary Figs. 1b, e; 2; 6; 7d, e were successfully repeated independently with identical results at minimum two times. The statistical significance for qPCR results was assessed using two-tailed *t*-test.

**Reporting summary**. Further information on research design is available in the Nature Research Reporting Summary linked to this article.

## Data availability
Datasets are available on NCBI Gene Expression Omnibus (Accession no. GSE135161). The mass spectrometry proteomics data have been deposited to the ProteomeXchange Consortium via the PRIDE partner repository with the dataset identifier PXD018373 and 10.6019/PXD018373. The source data underlying Figs. 1a, b, e, 2a–d, 3c, d, 4a, b, e, and 5a–e and Supplementary Figs. 1b, e, 2, 3e, 6, 7d, e are provided as a Source data file. All data is available from the corresponding author upon reasonable request.

## Code availability
All codes used in this study are publicly available.

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

## Acknowledgements

We thank R. Allshire for the *cwf10-1* mutant strain, members of the Grewal laboratory for valuable assistance and J. Barrowman for editing the paper. We also thank M. Zofall for suggestions and protocols, and V. Balachandran for strain constructions. This work was supported by the Intramural Research Program of the National Institutes of Health, National Cancer Institute. Bioinformatics analysis in this study used the Helix and Biowulf Linux cluster at the National Institutes of Health.

## Author contributions

S.I.S.G. and G.T. designed the study. G.T., H.X., S.H., and J.D. performed experiments. H.X. performed RIP-seq and ultracentrifugation experiments. S.H. performed sporulation frequency experiment. J.D. generated the *cbc-1* mutant. D.W. performed statistical analysis of RIME and provided bioinformatics support. L.M.J. performed mass spectrometry analysis; G.T. performed all other experiments and bioinformatics analyses. S.I.S.G. and G.T. prepared data figures. S.I.S.G. and G.T. wrote the paper.

## Competing interests

The authors declare no competing interests.
