## [Peer Review File · Nature Communications]

Reviewers' comments:

Reviewer #1 (Remarks to the Author):

In this paper, Thillainadesan and coworkers report a role for the conserved protein Pir2ARS2 in gene repression by cryptic intron-containing lncRNAs in fission yeast. Specifically, the authors show that i) Pir2 associates with lncRNAs and represses the expression of adjacent genes, ii) cryptic introns within lncRNAs control gene repression by Pir2 and splicing factors genomewide, iii) Pir2 interact with and recruit silencing effectors (RNAi, HDAC, FACT) to cryptic intron-containing lncRNAs, and iv) the human Pir2 homologue ARS2 associates with the splicing machinery.

Overall, the data are convincing and the concept of gene repression by cryptic intron-containing lncRNAs is novel and interesting. I think the manuscript is suitable for publication in Nature Communications, should additional controls/experiments be provided to further strengthen the model:

Major points:

- 1) It is unclear whether the reported interactions between Pir2 and CBC, splicing factors, RNAi, Clr3 and Pob3 are mediated or not by RNA. Do the physical interactions persist upon ribonuclease treatment?
- 2) Like *pir2-1*, mutants of *Cwf10* and *nam1* cryptic intron should restore sporulation defects observed in *rrp6Δ* cells (Fig 1f).
- 3) The model in Fig 2g and 3e would benefit from evidence showing a role for cryptic introns in recruiting Pir2, splicing factors and downstream effectors (e.g. by ChIP or RIP).
- 4) The MTREC components Pir2 and Red1 have strikingly different phenotypes with respect to the levels of lncRNAs and downstream genes (Fig 1a, 1b). I am wondering what would be the role of Mtl1, which associates with Red1 in MTREC or with splicing-associated factors (e.g. Nrl1) (Lee et al, 2013).

Minor points:

- 1) The effect of the *cbc-1* mutant on *pho1* expression is rather modest. Is it the same for additional targets? Does the mutant retain some kind of activity? RT-qPCR data would provide quantitative information about CBC contribution.
- 2) The Mtl1 cofactor Nrl1 interacts with *Cwf10* and promotes siRNA accumulation in *rrp6Δ* cells at retrotransposons and specific genes (Lee et al, 2013). What about the studied lncRNA loci?
- 3) in *pir2-1* cells, there should be less Ago1 recruited to lncRNA prt-L, such as observed for Clr3 and Pob3 (Fig 4e).
- 4) Legends for supplemental figure 1f and 1g are swapped.

Reviewer #2 (Remarks to the Author):

The manuscript by Thillainadesan et al describes interesting and well carried out work that elucidates a

pathway for nuclear RNA gene repression via the protein Pir2/ARS2 and long non-coding RNA (lncRNAs). Mutation in Pir2 upregulates genes transcribed downstream of lncRNAs suggesting that Pir2, along with other factors, are involved in repressing genes downstream of lncRNAs. A key result and proposal here is that Pir2 interacts with splicing factors and the presence of cryptic introns in some way mediates the repression. It is clear that mutation in Pir2 induces activation of cryptic intron splicing. However, the data currently presented does not provide a direct link to splicing factors in “active” spliceosomes and requires either additional experiments to prove a link to active spliceosomes or an alternative hypothesis that supports the current results.

It is an interesting observation that Pir2 associates with the splicing factor Cwf10. However, the statements “Pir2-purified fractions also contained a subset of splicing factors” and “Our biochemical analyses showed that Pir2 indeed forms a complex with splicing factors.” on page 3, paragraph 3, are misleading as only data for Cwf10 has been shown here, not any other splicing factors. Either the data supporting Pir2 interaction with other splicing factors should be included or this text should be changed.

Glycerol gradient analysis of Pir2/ARS2 associating with a complex containing Cwf10/EFTUD2 shown in Figures 2b and Supplementary Figure 5d needs additional information and better annotation. In glycerol gradient analysis of splicing complexes, different snRNP complexes are found in different parts of the gradient. Ideally, a northern blot for all the snRNAs (or at least U2, U4, U5 and U6) showing where each snRNA is and, therefore, where each splicing complex runs should be included here to show the reader where each of the splicing complexes are found in the gradient. The free U5 snRNP, U4/U6 snRNP and the active spliceosome U2/U5/U6 snRNP complex should be annotated on the gradient from information obtained by a northern blot of the snRNAs from the same fractions. This information is essential, as from the data presented it appears that Pir2/ARS2 only associates with U5 snRNPs in the early, smaller complex containing, fractions of the gradients. Importantly, these smaller, U5 snRNP only, complexes would not be actively engaged in splicing and therefore would not support the association of Pir2/ARS2 with active spliceosomes.

Indeed, the mass spectrometry data presented in Supplementary 5a clearly indicate that ARS2 is pulling down early, non-activated spliceosomes, as pre-catalytic splicing factors like DDX23/Prp28, PRPF3/Prp3 and SART1/Snu66 are still present (these proteins are removed in remodelling to the active spliceosome) and no catalytic spliceosome-specific proteins, like those of the nineteen complex (NTC), are present. Therefore, an alternative hypothesis is that Pir2/ARS2 is required during the assembly of the activated spliceosome but is not present in the active spliceosome directly interacting with normal or cryptic introns. The activation of cryptic introns upon Pir2 mutation, therefore, may be a consequence of reduced/defective spliceosome assembly and not a defect in Pir2 associating with active splicing complexes on RNA.

An important experiment the authors can easily do is to look at the RNA components of their immunoprecipitations of Pir2 to see what snRNAs Pir2 is associating with. This experiment will back up results from the glycerol gradient analysis and give an indication of the splicing complexes that Pir2 associates with.

Following these changes the authors need to present their results within the context of what is known about how splicing complexes run in glycerol gradients and the known pathway of spliceosome assembly and splicing factor association with these complexes, as exemplified by recent cryo-EM analysis of the spliceosome carried out in humans and *S. pombe* by the Shi, Luhrmann and Nagai labs. At present the data supports Pir2 interacting only with early splicing complexes.

Other issues to address.

- 1) The following sentence in the Abstract is not clear: “We find that invasion of a lncRNA into the neighboring gene results in inclusion of a cryptic intron that is required for its repressive activity.” This sentence would read more clearly if it was modified to: “We find that invasion of lncRNA transcription into a neighboring gene results in inclusion of a cryptic intron that is required for repression of that neighboring gene.”
- 2) The following sentence in the Abstract is not clear: “Pir2 and splicing machinery are broadly required for repression genome-wide.” This sentence would read more clearly if it was modified to: “Pir2 and splicing machinery are broadly required for gene repression genome-wide.”
- 3) Supplementary Fig. 1f and Supplementary Fig. 1g are mixed up in the second paragraph on page 3.
- 4) Pir2 association with the RNA by RNA immunoprecipitation in Figure 1d, please explain why association is sometimes within the coding gene and sometimes it is not?
- 5) Ideally experiment in Figure 1e should be repeated with another lncRNA deletion like Nam1.
- 6) RT-PCR should be used to validate cryptic intron inclusion from the RNA-seq data.
- 7) Percentages shown in Figure 2e and Supplementary Fig 2d should be clearly explained in the text as well as the legends. The statement found in Figure 2e legend only: “Numbers indicate percentage of reads spliced.” is not really clear. It does not give a clear indication of how prevalent the cryptic splice is which appears to be very low. So does 3% mean that only 3% of messages have a cryptic splice and 97% of messages don't? It would also be useful to add the exact numbers of cryptic exon events detected by RNA-seq. For example, if 3% is only 3 events in 10 then is this number above or below the threshold of confidence one would have for the event?
- 8) A negative control for intron mutations would be useful for the experiment in Figure 2F. For example, can you make a mutation within the intron that does not affect splice sites and therefore does not increase Pho1 expression?

Reviewer #3 (Remarks to the Author):

This manuscript by Thillainadesan et al. describes how lncRNA acts to repress the neighboring gene through Pir2 (ARS2 in mammals) and splicing factors. Overall, the study is well conducted and will be important to the field. However, this manuscript presents several correlation analyses, which should be appreciated. My points are listed below:

1. Fig1a and b: The authors should add probes on prt and nam1 lncRNA genes for northern blot analysis. This is an important control experiment to show a termination defect in mmi, rrp6 and red1-deletion mutants, but not in pir2 mutant (pir2-1). Short isoforms of prt and nam1 should be detected in WT and pir2-1 by the lncRNA probes, although levels of long isoforms (L) should be increased in other mutants?
2. Fig1e: The authors should deplete prt lncRNA using siRNA technology to discuss lncRNA function. This knockdown experiment should not show additive effect on pho1 expression as well as deletion of prt lncRNA region?
3. Fig2 and Sup Fig5: Why did the authors focus on Cwf10 (EFTUD2 in human) only? They should show other splicing factors such as Cwf21(SRRM2), Spp42 (PRPF8) and Prp10 (SF3B1). According to Supplementary Figure 5, ARS2 is preferentially associated with those catalytic spliceosome components in human cells. The authors should comment on how “catalytic” spliceosome (not just splicing factors) acts as this gene suppression.
4. Fig2d: Primer sets of prt and nam1 for qRT-PCR should be tested to show lncRNA levels.
5. Fig2g: This is a very nice experiment. However, addition of the same approach onto byr2 gene will make this stronger. Splicing inhibitor such as Pladienolide B or SSA that affect catalytic spliceosome formation should also be examined. Furthermore I am interested to see if combination of the intron mutations and pir2-1 induces an additive effect on pho1 expression.
6. Sup Fig2c: Profile of cryptic-intron less genes should be addressed as control.
7. Sup Fig2 d-f: Few more retrotransposon examples should be appreciated.
8. Fig3a and b: These interactions should be RNA-dependent if lncRNA mediates gene repression. RNase treatment should be tested in the IP experiments.
9. Sup Fig3b: The same Y-scale should be used to emphasize the huge differences.

Response to reviewers' comments:

We are most grateful to the reviewers for their valuable comments. We have extensively revised the paper and have included new results, including analysis of additional splicing factors and the requested control experiments. Our new analyses address the concerns raised by the reviewers and further strengthen the main conclusions of our work. We believe that our study is considerably improved and now meets the standards of quality and novelty expected for publication in *Nature Communications*.

Below is a point-by-point response to the reviewers' comments (indicated in blue text).

Reviewer #1 (Remarks to the Author):

In this paper, Thillainadesan and coworkers report a role for the conserved protein Pir2/ARS2 in gene repression by cryptic intron-containing lncRNAs in fission yeast. Specifically, the authors show that i) Pir2 associates with lncRNAs and represses the expression of adjacent genes, ii) cryptic introns within lncRNAs control gene repression by Pir2 and splicing factors genomewide, iii) Pir2 interact with and recruit silencing effectors (RNAi, HDAC, FACT) to cryptic intron-containing lncRNAs, and iv) the human Pir2 homologue ARS2 associates with the splicing machinery. Overall, the data are convincing and the concept of gene repression by cryptic intron-containing lncRNAs is novel and interesting. I think the manuscript is suitable for publication in Nature Communications, should additional controls/experiments be provided to further strengthen the model:

We thank the reviewer for the positive comments. We have included additional control experiments as requested and addressed the specific points raised. Please see below for our point-by-point response.

Major points:

1) It is unclear whether the reported interactions between Pir2 and CBC, splicing factors, RNAi, Clr3 and Pob3 are mediated or not by RNA. Do the physical interactions persist upon ribonuclease treatment?

We now include controls in the revised Supplementary Fig. 6 to address this question. CBC and Pir2 were previously reported to interact as a stable complex independent of RNA (PMID: 25989903). We conducted Benzonase treatment to test all of the other interactions and found that most were not abolished by Benzonase treatment, suggesting that the interactions are RNA independent. However, the Pob3 and Pir2 interaction appears to be partially sensitive to Benzonase treatment, suggesting that RNA may have a role in promoting co-transcriptional association of Pob3 with Pir2.

2) Like pir2-1, mutants of Cwf10 and nam1 cryptic intron should restore sporulation defects observed in rrp6Δ cells (Fig 1f).

We thank the reviewer for this excellent suggestion. In the revised Fig. 2e we show that a mutation in *cwf10* (*cwf10-1*) was indeed able to rescue the sporulation defect of the *rrp6Δ* single mutant. However, mutation of the cryptic intron in the *nam1* lncRNA requires amino acid changes that will likely disrupt the function of the Byr2 protein and result in some degree of sporulation defect. For this reason, it would be difficult to draw a conclusion from this experiment. Furthermore, construction and characterization of strains required for this experiment are time-consuming and could not be completed within the time frame allotted for revisions.

3) The model in Fig 2g and 3e would benefit from evidence showing a role for cryptic introns in recruiting Pir2, splicing factors and downstream effectors (e.g. by ChIP or RIP).

Our model suggests that Pir2 docks at the lncRNA along with CBC as well as Mmi1-MTREC. The cryptic intron, which is incorporated by transcriptional invasion into the coding region of the downstream gene, helps connect Pir2 to the splicing machinery. In turn, Pir2 and splicing factors act together to engage silencing effectors and repress target gene expression. As per this model, deletion of the cryptic intron would not abolish the localization of Pir2 since other factors such as CBC and MTREC that associate with Pir2 bind to lncRNA independently of the cryptic intron. Consistent with this prediction, the *cwf10-1* mutant did not abolish Pir2-GFP occupancy at the *pho1* locus.

4) The MTREC components Pir2 and Red1 have strikingly different phenotypes with respect to the levels of lncRNAs and downstream genes (Fig 1a, 1b). I am wondering what would be the role of Mtl1, which associates with Red1 in MTREC or with splicing-associated factors (e.g. Nrl1) (Lee et al, 2013).

Red1 and Mtl1 form the core of MTREC. We show that deletion of Red1 results in the stabilization of the lncRNA. Pir2, which is a factor that associates with MTREC and the splicing machinery, is primarily responsible for promoting repression of the downstream gene following association with the splicing factors through the cryptic intron. Considering our previous work showed that Mtl1 also forms a Red1-independent complex with Nrl1 and its associated splicing factors (PMID: 24210919), the reviewer wonders whether Mtl1 and/or Nrl1 would have a role in controlling the levels of lncRNA or the silencing of downstream genes. As suggested, we have performed additional experiments to follow up on this excellent question. Since Mtl1 is an essential gene, we used a temperature-sensitive allele to assess its role at the *prt-pho1* locus. Our northern blot analysis shows that in the *mtl1-1* mutant, the *prt* lncRNA is stabilized (see figure →), similar to the result in *red1Δ* (see Fig. 1a). This is consistent with the fact that Red1 and Mtl1 are core components of MTREC. Additionally, *nrl1Δ* caused only little or no change in *pho1* or lncRNA expression. This result indicates

Northern blot analysis of transcripts produced from the *pho1* locus. The black line indicates the position of the radioactive probe. Ribosomal RNA was used as a loading control.

that Nrl1 is dispensable for gene repression mediated by Pir2 and splicing machinery via the lncRNA, although we cannot rule out the possibility that redundant factors mask the effects of *nrl1* Δ .

Minor points:

1) *The effect of the cbc-1 mutant on pho1 expression is rather modest. Is it the same for additional targets? Does the mutant retain some kind of activity? RT-qPCR data would provide quantitative information about CBC contribution.*

We conducted RT-qPCR analysis of the *pho1*, *tgp1* and *byr2* genes in strains that carried the *cbc1-1* mutant and found that there was an increase in gene expression in the mutants as compared to the WT (see figure \rightarrow). The difference in expression is less than observed in the *pir2-1* mutant. We think it is likely that compared to the *pir2-1* mutant, *cbc1-1* only weakly affects Cbc1 function at the restrictive temperature. Notably, since the full deletion causes lethality, temperature-sensitive mutations yield only a partial phenotype that may vary in strength between mutants.

RT-qPCR analysis of loci controlled by lncRNAs in wild-type (WT) and *cbc1-1* mutant cells. Transcript levels were normalized to the *leu1* control. Gene specific primers were used for amplification. Each bar represents mean values \pm SD for $n \geq 3$.

2) *The Mtl1 cofactor Nrl1 interacts with Cwf10 and promotes siRNA accumulation in rrp6 Δ cells at retrotransposons and specific genes (Lee et al, 2013). What about the studied lncRNA loci?*

Our lab previously reported that Nrl1 aids in the production of siRNAs at retrotransposons and some specific genes (PMID: 24210919). However, loss of Nrl1 has no effect at the lncRNA controlled loci such as *pho1* (PMID: 24210919). Additionally, as mentioned above our northern blot results show that loss of Nrl1 has no major effect on *pho1* gene expression.

3) *in pir2-1 cells, there should be less Ago1 recruited to lncRNA prt-L, such as observed for Clr3 and Pob3 (Fig 4e).*

Ago1 is a very challenging protein to ChIP. For this reason, it is difficult to conclusively determine the localization of Ago1 in various mutants. However, our previous work showed that Ago1 is essential for siRNA production at lncRNAs (PMID: 23151475), and our northern blot in Fig. 3e shows that loss of Ago1 causes a drastic upregulation of the *pho1* gene. Additionally, Pir2 interacts with Hrr1, a component of the RNA-dependent RNA polymerase complex, and our IP results demonstrate that in the *cwf10-1* mutant, the interaction between Hrr1 and Pir2 is strongly diminished (Fig. 3b). This suggests that the recruitment of RNAi factors to loci that are dependent on Pir2 and Cwf10, such as *pho1* and *byr2*, is likely affected in *pir2-1* cells.

4) *Legends for supplemental figure 1f and 1g are swapped.*

We thank the reviewer for noting this. The legends have been fixed.

Reviewer #2 (Remarks to the Author):

The manuscript by Thillainadesan et al describes interesting and well carried out work that elucidates a pathway for nuclear RNA gene repression via the protein Pir2/ARS2 and long non-coding RNA (lncRNAs). Mutation in Pir2 upregulates genes transcribed downstream of lncRNAs suggesting that Pir2, along with other factors, are involved in repressing genes downstream of lncRNAs. A key result and proposal here is that Pir2 interacts with splicing factors and the presence of cryptic introns in some way mediates the repression. It is clear that mutation in Pir2 induces activation of cryptic intron splicing. However, the data currently presented does not provide a direct link to splicing factors in "active" spliceosomes and requires either additional experiments to prove a link to active spliceosomes or an alternative hypothesis that supports the current results.

It is an interesting observation that Pir2 associates with the splicing factor Cwf10. However, the statements "Pir2-purified fractions also contained a subset of splicing factors" and "Our biochemical analyses showed that Pir2 indeed forms a complex with splicing factors." on page 3, paragraph 3, are misleading as only data for Cwf10 has been shown here, not any other splicing factors. Either the data supporting Pir2 interaction with other splicing factors should be included or this text should be changed.

Glycerol gradient analysis of Pir2/ARS2 associating with a complex containing Cwf10/EFTUD2 shown in Figures 2b and Supplementary Figure 5d needs additional information and better annotation. In glycerol gradient analysis of splicing complexes, different snRNP complexes are found in different parts of the gradient. Ideally, a northern blot for all the snRNAs (or at least U2, U4, U5 and U6) showing where each snRNA is and, therefore, where each splicing complex runs should be included here to show the reader where each of the splicing complexes are found in the gradient. The free U5 snRNP, U4/U6 snRNP and the active spliceosome U2/U5/U6 snRNP complex should be annotated on the gradient from information obtained by a northern blot of the snRNAs from the same fractions. This information is essential, as from the data presented it appears that Pir2/ARS2 only associates with U5 snRNPs in the early, smaller complex containing, fractions of the gradients. Importantly, these smaller, U5 snRNP only, complexes would not be actively engaged in splicing and therefore would not support the association of Pir2/ARS2 with active spliceosomes.

Indeed, the mass spectrometry data presented in Supplementary 5a clearly indicate that ARS2 is pulling down early, non-activated spliceosomes, as pre-catalytic splicing factors like DDX23/Prp28, PRPF3/Prp3 and SART1/Snu66 are still present (these proteins are removed in remodelling to the active spliceosome) and no catalytic spliceosome-specific proteins, like those of the nineteen complex (NTC), are present. Therefore, an alternative hypothesis is that Pir2/ARS2 is required during the assembly of the activated spliceosome but is not present in the active spliceosome directly interacting with normal or cryptic introns. The activation of cryptic introns upon Pir2 mutation, therefore, may be a consequence of reduced/defective spliceosome assembly and not a defect in Pir2 associating with active splicing complexes on RNA.

An important experiment the authors can easily do is to look at the RNA components of their immunoprecipitations of Pir2 to see what snRNAs Pir2 is associating with. This experiment will back up results from the glycerol gradient analysis and give an indication of the splicing

complexes that Pir2 associates with.

*Following these changes the authors need to present their results within the context of what is known about how splicing complexes run in glycerol gradients and the known pathway of spliceosome assembly and splicing factor association with these complexes, as exemplified by recent cryo-EM analysis of the spliceosome carried out in humans and *S. pombe* by the Shi, Luhrmann and Nagai labs. At present the data supports Pir2 interacting only with early splicing complexes.*

We thank the reviewer for the positive comments. The reviewer requested that we distinguish splicing factors migrating along with Pir2/ARS2 from the active spliceosome complex. We have conducted further experiments to address the reviewer's specific points as described below.

The reviewer recommended looking at the RNA components of the Pir2 immunoprecipitation to determine which snRNAs are present, and then using this information to determine if it is a pre-active or active spliceosome. However, our RIP-seq data of Pir2 shows enrichment at snRNAs since Pir2/ARS2 has been demonstrated to be involved in the proper processing and maturation of snRNAs (PMID:24270878)). For this reason, it is difficult to draw any conclusion using this method.

We instead addressed this point in another way, by conducting glycerol gradient analysis of additional splicing factors (revised Supplementary Fig. 2). To distinguish the elution pattern of the activated NTC spliceosome complex, we used Cdc5, a core component of the NTC complex and the activated spliceosome, as a marker. We show that Cdc5 elutes as part of a larger complex that is distinct from the elution peak of Pir2/ARS2. We also tested another spliceosome protein, Spp42 (Prp8), which stoichiometrically purifies with Cdc5 as part of the active spliceosome (PMID: 24442611). This protein also co-migrates as a large complex, similar to Cdc5. This suggests that the subset of splicing factors that elutes with Pir2 is not part of the activated spliceosome. Indeed, we were also able to identify an additional splicing factor, Cwf21 (SRRM2-humans) that co-migrates with Pir2. This protein is also present in the Pir2 purifications in *S. pombe* as well as in our hARS2 purification reported in this study. Additionally, although Cwf21 has been shown to associate with the spliceosome in some cases, there is evidence that RSR-2 (the *C. elegans* homologue of Cwf21) is primarily involved in regulating transcript expression levels (PMID: 23754964). The association of Pir2 with this subcomplex of splicing factors may allow recognition of the intronic sites in the downstream gene and provide a scaffold for the binding of chromatin modifying and RNA degradation factors. Indeed, we demonstrate in Fig. 3b that association of Pir2 with Hrr1 is abolished when the splicing factor Cwf10 is mutated. The active splicing event is likely dispensable for the repression.

Interestingly, the splicing of cryptic introns is mainly detected in the absence of RNA processing factors, such as the exosome subunit Rrp6 and the Pir2 that recruits RNAi machinery to degrade target transcripts. This is most likely because of kinetic competition between RNA processing and splicing machineries. In cells defective in RNA processing, the balance is shifted to splicing, ultimately leading to detectable levels of splicing events.

In sum, our analyses suggest that Pir2 does not associate with the active spliceosome. Instead, Pir2 acts together with a subset of splicing factors that have been co-opted to engage silencing effectors including RNA degradation and chromatin modifying factors, independently of splicing activity. In other words, the association of Pir2 with splicing factors, rather than the act of splicing per se, is important for lncRNA-mediated gene repression. We have modified the text to address the points discussed above.

Other issues to address.

1) *The following sentence in the Abstract is not clear: "We find that invasion of a lncRNA into the neighboring gene results in inclusion of a cryptic intron that is required for its repressive activity." This sentence would read more clearly if it was modified to: "We find that invasion of lncRNA transcription into a neighboring gene results in inclusion of a cryptic intron that is required for repression of that neighboring gene."*

We have modified the sentence to make the meaning clearer.

2) *The following sentence in the Abstract is not clear: "Pir2 and splicing machinery are broadly required for repression genome-wide." This sentence would read more clearly if it was modified to: "Pir2 and splicing machinery are broadly required for gene repression genome-wide."*

We have modified the sentence.

3) *Supplementary Fig. 1f and Supplementary Fig. 1g are mixed up in the second paragraph on page 3.*

We thank the reviewer for pointing out this error and we have made the correction.

4) *Pir2 association with the RNA by RNA immunoprecipitation in Figure 1d, please explain why association is sometimes within the coding gene and sometimes it is not?*

In Fig. 1d, the results of Pir2 enrichment as detected by RNA immunoprecipitation reflects the abundance and length of the lncRNA at the loci shown.

5) *Ideally experiment in Figure 1e should be repeated with another lncRNA deletion like Nam1.*

It has been previously demonstrated that placing a terminator sequence at *nam1* promotes upregulation of *byr2* (PMID: 28765164), similar to the effect of deleting *pvt* on *pho1* transcript levels. Our northern blot (Fig 1a) shows the remarkable similarity of the effect on *pho1* and *byr2* loci in *mmi1Δ*, *rrp6Δ* and *red1Δ*. Finally, our RIP-seq and ChIP-seq data demonstrate that Pir2 clearly localizes to the *nam1* and *pvt* ncRNA just 5' to the gene. Considering our results that Pir2 localizes to the ncRNA along with the MTREC complex, we believe that deletion of *nam1*, much like deletion of *pvt*, would abolish Pir2 occupancy at the locus.

6) *RT-PCR should be used to validate cryptic intron inclusion from the RNA-seq data.*

The cryptic introns are known to be spliced very inefficiently, which makes it technically challenging to detect these introns using conventional RT-PCR. Due to the

very low level of splicing in the *pir2-1* and *cwf10-1* mutants, splicing events can only be detected using a sequencing approach. The added advantage of the sequencing method is that it allows verification of splice junctions that map to consensus splice sites.

7) Percentages shown in Figure 2e and Supplementary Fig 2d should be clearly explained in the text as well as the legends. The statement found in Figure 2e legend only: "Numbers indicate percentage of reads spliced." Is not really clear. It does not give a clear indication of how prevalent the cryptic splice is which appears to be very low. So does 3% mean that only 3% of messages have a cryptic splice and 97% of messages don't? It would also be useful to add the exact numbers of cryptic exon events detected by RNA-seq. For example, if 3% is only 3 events in 10 then is this number above or below the threshold of confidence one would have for the event?

The percentages were calculated for reads that overlap with the intronic region, since this is an accurate representation of splicing events that occur. Because RNA-seq library preparation is subject to bias within a locus, certain regions are amplified better than others. In short, the number of spliced reads were taken as a percentage of the total number of reads overlapping with the intron. Thus, 3% means that 3 reads out of the 100 reads overlapping in that region are split reads. We understand that further explanation is needed, so we have added a more comprehensive description of this calculation in the materials and methods under the 'Bioinformatics' heading.

*8) A negative control for intron mutations would be useful for the experiment in Figure 2F. For example, can you make a mutation within the intron that does not affect splice sites and therefore does not increase *Pho1* expression?*

This is a very interesting suggestion, however the strain construction for this experiment is extremely challenging and time consuming. We were not able to obtain the strains within the time frame allotted for the revision. In any case, we are confident of the results presented in the paper that were confirmed using two independently constructed strains. In each case, mutation of the 5' and 3' splice sites lead to de-repression of the target locus.

Reviewer #3 (Remarks to the Author):

This manuscript by Thillainadesan et al. describes how lncRNA acts to repress the neighboring gene through Pir2 (ARS2 in mammals) and splicing factors. Overall, the study is well conducted and will be important to the field. However, this manuscript presents several correlation analyses, which should be appreciated. My points are listed below:

We thank the reviewer for the positive feedback and provide a point-by-point response below.

*1. Fig1a and b: The authors should add probes on *pvt* and *nam1* lncRNA genes for northern blot analysis. This is an important control experiment to show a termination defect in *mmi*, *rrp6* and *red1*-deletion mutants, but not in *pir2* mutant (*pir2-1*). Short isoforms of *pvt* and *nam1* should be detected in WT and *pir2-1* by the lncRNA probes, although levels of long isoforms (L) should be increased in other mutants?*

We have conducted northern blot analysis using a probe specific for the *nam1* ncRNA. We were only able to detect small amounts of *nam1* short RNA in WT (see figure →). This is consistent with our RNA-seq data that also showed very few reads at the ncRNA region. We only detect enrichment for this region in the form of the long versions of the ncRNAs (*pvt-L* or *nam1-L*) or in our Pir2 RIP-seq data after Pir2 has been crosslinked and enriched. The reason why the short ncRNAs are not detected is that as we and others have previously shown that ncRNA is bound by Mmi1 and its associated RNA degradation (e.g. MTREC and Rrp6) and RNAPII termination (cleavage and polyadenylation factor, CPF) factors, which promote transcription termination and immediate degradation of the RNA by Rrp6 (PMID: 31269446). In any case, defects in termination of lncRNA in *mmi1Δ* and *rrp6Δ* have been also documented previously (PMID: 24493644; 28765164).

Northern blot analysis of transcripts produced from the *nam1-byr2* locus. The black line indicates the position of the radioactive probe. Ribosomal RNA was used as a loading control.

2. Fig1e: The authors should deplete *pvt* lncRNA using siRNA technology to discuss lncRNA function. This knockdown experiment should not show additive effect on *pho1* expression as well as deletion of *pvt* lncRNA region?

Unfortunately, siRNA knockdowns cannot be performed in *S. pombe*.

3. Fig2 and Sup Fig5: Why did the authors focus on *Cwf10* (EFTUD2 in human) only? They should show other splicing factors such as *Cwf21* (SRRM2), *Spp42* (PRPF8) and *Prp10* (SF3B1). According to Supplementary Figure 5, ARS2 is preferentially associated with those catalytic spliceosome components in human cells. The authors should comment on how "catalytic" spliceosome (not just splicing factors) acts as this gene suppression.

We have now tested *Cwf21*, *Spp42* and *Cdc5* using glycerol gradients and have improved the annotation of these in the revised Supplementary Fig. 2. Interestingly, *Cwf21* co-migrates with *Pir2*/ARS2. *Cwf21* has been shown to associate with the spliceosome, but there is evidence that RSR-2 (*C. elegans* homologue of *Cwf21*) is primarily involved in regulating the expression level of transcripts (PMID: 23754964). This suggests that *Pir2* co-migrates and associates with a subset of splicing factors (*cwf10* and *cwf21*), which might also participate in gene repression. However, *Spp42* and *Cdc5* migrate as large complexes, which represent the activated spliceosome complex (PMID: 24442611) but are absent in the fractions containing *Pir2*. These results suggest that the *Pir2* is not associated with the active spliceosome. Based on these analyses, we envision that a subset of splicing factors has been co-opted to mediate gene repression by *Pir2*/ARS2 and its associated effectors. In other words, the association of *Pir2* with splicing factors, rather than the act of splicing per se, is important for lncRNA-mediated gene repression.

4. Fig2d: Primer sets of *pvt* and *nam1* for qRT-PCR should be tested to show lncRNA levels.

The northern blots depicted in Fig. 1b and Fig. 2c use a probe that can detect both the gene (the band indicated for *pho1* and *byr2*) and the lncRNA that runs through the gene (the band indicated as *pvt-L* and *nam1-L*). Our analyses show that cells carrying the *pir2-1* and *cwf10-1* alleles do not show accumulation of the long ncRNA, in contrast to the effect of loss of Mmi1, Red1 or Rrp6 that control the stability of lncRNA (Fig. 1a). Rather, Pir2 and Cwf10, which bind to lncRNA, mediate repression of downstream genes. Considering the results of our northern blot analyses, no major changes in the levels of *pvt* and *nam1* RNA are expected in *pir2-1* or *cwf10-1* mutant cells.

5. Fig2g: This is a very nice experiment. However, addition of the same approach onto *byr2* gene will make this stronger. Splicing inhibitor such as Pladienolide B or SSA that affect catalytic spliceosome formation should also be examined. Furthermore I am interested to see if combination of the intron mutations and *pir2-1* induces an additive effect on *pho1* expression.

Based on our glycerol gradient analyses, we don't believe that the actual catalytic spliceosome is responsible for the repression. Inhibiting the formation of the large catalytic spliceosome will most likely lead to defective splicing and is expected to cause pleiotropic changes. As recommended, we have tested the epistatic nature and found that the double *pir2-1* and cryptic intron mutant does not show a cumulative effect on the *pho1* gene expression level, suggesting that Pir2 works through the cryptic intron to promote gene repression (revised Fig. 3d).

6. Sup Fig2c: Profile of cryptic-intron less genes should be addressed as control.

For this figure, we evaluated only the genes that were upregulated in the *pir2-1* mutant to determine if *cwf10-1* also had a similar or greater cumulative effect on these genes. Cryptic-intron less genes would not be a useful control for this figure.

7. Sup Fig2 d-f: Few more retrotransposon examples should be appreciated.

The *S. pombe* genome only contains 13 full length copies of Tf2 retrotransposon. Tf2 distributed across three chromosomes are identical, hence it is not possible to differentiate between different copies of retrotransposons. In other words, all transposable elements will show the same expression profile. Thus, depicting them does not provide any additional value.

8. Fig3a and b: These interactions should be RNA-dependent if lncRNA mediates gene repression. RNase treatment should be tested in the IP experiments.

We conducted immunoprecipitation experiments in the presence of Benzonase. We find that most of the Pir2-interacting factors that we present in this study are not sensitive to Benzonase treatment, with the exception of Pob3 (revised Supplementary Fig. 6). Our results suggest that Pir2 in general associates with silencing effectors through protein-protein interactions, and the lncRNA provides a scaffold for the recruitment of Pir2 and its associated factors. However, RNA may have a role in promoting co-transcriptional association of Pob3 with Pir2. We have included these results and have modified the text accordingly.

9. *Sup Fig3b: The same Y-scale should be used to emphasize the huge differences.*

As recommended, we have modified the figure panels and have used same Y-scale to emphasize the differences in the levels of small RNAs (see modified Supplementary Fig. 4). We have included an additional panel with a different scale to show that the siRNA size distribution is affected in the *pir2-1* and *cwf10-1* mutants as compared to *rrp6Δ*.

Reviewers' comments:

Reviewer #1 (Remarks to the Author):

In the revised version of the manuscript, the authors have answered to my initial comments. I recommend publication.

Reviewer #2 (Remarks to the Author):

The revised manuscript by Thillainadesan et al now provides convincing additional experiments and a revised interpretation of their results that now address my previous concerns. It is clear from the new data presented in Supplementary Figure 2 that Pir2 forms a complex with splicing factors that are not part of the active spliceosome. The authors now present/discuss these data clearly in the Results and Discussion providing a correct interpretation of these interesting results. The authors have also addressed all my other points and concerns sufficiently. The only minor thing that should be added to the Materials and Methods are the details of the antibodies used for western blotting for Cdc5 and Spp42(Prp8).

Reviewer #3 (Remarks to the Author):

Thillainadesan et al., observed interesting gene regulation using Pir2 mutant. They have improved their manuscript based on the reviewers' comments, but not significantly. Importantly I cannot agree with their working model.

I saw their response to one of Reviewer #2 comment, which is about UsnRNAs. They answered that it is difficult to check the UsnRNAs, since Pir2/ARS is involved in UsnRNA maturation (PMID: 24270878). This is critically important for their model. In fact, they don't get rid of a possibility which is an indirect effect of Pir2 on silencing of gene that contains a cryptic intron.

I also don't think that they properly did IP experiments to see protein-protein interaction in chromatin. Their nuclear lysis condition is too weak (at least in mammalian cells). Most likely they analyzed nuclear fraction that does not contain chromatin-associated lncRNAs and RNA binding proteins which they want to analyze. They must have shown evidence if they used proper chromatin fraction for the IP. They should isolate and solubilize chromatin to perform the IP experiments and then perform RNase treatment to examine the (lnc)RNA dependency.

Response to reviewers' comments:

Reviewer #1

In the revised version of the manuscript, the authors have answered to my initial comments. I recommend publication.

We thank the reviewer for supporting publication of our work.

Reviewer #2:

The revised manuscript by Thillainadesan et al now provides convincing additional experiments and a revised interpretation of their results that now address my previous concerns. It is clear from the new data presented in Supplementary Figure 2 that Pir2 forms a complex with splicing factors that are not part of the active spliceosome. The authors now present/discuss these data clearly in the Results and Discussion providing a correct interpretation of these interesting results. The authors have also addressed all my other points and concerns sufficiently. The only minor thing that should be added to the Materials and Methods are the details of the antibodies used for western blotting for Cdc5 and Spp42(Prp8).

We thank the reviewer for supporting publication of our work. As recommended, we have added details describing the antibodies used for detection of Cdc5-HA and Spp42-HA to the Methods section.

Reviewer #3:

Thillainadesan et al., observed interesting gene regulation using Pir2 mutant. They have improved their manuscript based on the reviewers' comments, but not significantly. Importantly I cannot agree with their working model.

I saw their response to one of Reviewer #2 comment, which is about UsnRNAs. They answered that it is difficult to check the UsnRNAs, since Pir2/ARS is involved in UsnRNA maturation (PMID: 24270878). This is critically important for their model. In fact, they don't get rid of a possibility which is an indirect effect of Pir2 on silencing of gene that contains a cryptic intron.

Reviewer 2 suggested examining snRNA components to determine whether Pir2 associates with factors that are part of the active spliceosome. However, since our RNA immunoprecipitation and sequencing (RIP-seq) analyses show that Pir2 binds to snRNAs, which is consistent with involvement of Pir2/ARS2 in processing of snRNAs, we used an alternative strategy to address this issue. Specifically, our glycerol gradient analyses of additional splicing factors clearly show that Pir2 forms a complex with splicing factors that are not part of the active spliceosome (Supplementary Fig. 2). Based on the comments of Reviewer 2 above, the new results that we have included in the paper address his/her concerns.

We do not understand the comment that the effect of Pir2 on silencing might be indirect. Presumably, the reviewer is suggesting that defective silencing is due to general changes in splicing. However, *pir2-1* mutant cells do not show general defects in splicing, but rather show increased splicing of cryptic introns in lncRNAs (see Fig. 3a). Strongly supporting the direct involvement of Pir2, we show that Pir2 localizes to target loci that are silenced by the lncRNAs containing cryptic introns (see Fig. 1c, d). Moreover, the localization of Pir2 at target loci such as the *pho1* gene requires lncRNA (Fig. 1e). Also, as determined by our epistasis analysis, the double mutant carrying both *pir2-1* and the cryptic intron mutations does not show a cumulative effect on levels of *pho1* gene expression as compared to the single mutants (Fig. 3d). Together, these results strongly argue in favor of Pir2 directly mediating the silencing effect of lncRNAs on downstream target genes.

I also don't think that they properly did IP experiments to see protein-protein interaction in chromatin. Their nuclear lysis condition is too weak (at least in mammalian cells). Most likely they analyzed nuclear fraction that does not contain chromatin-associated lncRNAs and RNA binding proteins which they want to analyze. They must have shown evidence if they used proper chromatin fraction for the IP. They should isolate and solubilize chromatin to perform the IP experiments and then perform RNase treatment to examine the (lnc)RNA dependency.

The reviewer is presumably implying that the interactions between Pir2 and other factors would only be detected in the chromatin-associated fraction that contains RNA binding proteins. However, please note that our experiments were performed using an established protocol and successfully detected interactions between Pir2 and silencing factors that we show are required for lncRNA-mediated gene repression. As suggested by other reviewers, we included a Benzonase treatment step to test whether the interactions between Pir2 and its partners, including the splicing machinery and the effector proteins, are RNA dependent. Our result suggests that these complexes can assemble independently of RNA (Supplementary Fig. 6). Indeed, our ChIP experiments show that the Pir2 mutation disrupts the localization of the effector proteins at the target locus (Fig 5e). Moreover, epistasis analyses reveal that silencing effectors act together with Pir2 (Fig. 5d). For RIME experiment, our method yields proper and sufficient nuclear lysis of HepG2 cells for the following reasons. First, the experiment was performed as previously reported (PMID:26797456). The lysis condition used in this study is well characterized and the buffers are commonly used for lysis in other assays such as chromatin immunoprecipitation experiments (PMID: 22113270). The nuclei were first isolated using a nuclear extraction buffer and ultimately lysed using detergent strong enough to lyse the nuclear membrane. Second, the lysate was subjected to sonication, which further promotes lysis and shearing of chromatin. We confirmed optimal sonication and solubilization of chromatin by isolating chromatin-associated DNA prior to the addition of the antibody. This procedure confirms the presence of solubilized chromatin and indicates that proper lysis of the nuclei was achieved. We have revised the description of immunoprecipitation experiments to further clarify procedures used in our experiments.